# Role of Nitric Oxide and Nrf2 to Counteract Vascular Endothelial Dysfunction Induced by Periodontal Pathogens Using HUVECs

**DOI:** 10.3390/cells14221777

**Published:** 2025-11-12

**Authors:** Gunaraj Dhungana, Chethan Sampath, Vineeta Sharma, Olga Korolkova, Pandu R. Gangula

**Affiliations:** 1Department of Oral Diagnostic Sciences & Research, School of Dentistry, Meharry Medical College, Nashville, TN 37208, USA; gunaraj.dhungana1@mmc.edu (G.D.); csampath@mmc.edu (C.S.); 2Department of Biochemistry, Cancer Biology, Neuroscience and Pharmacology, Meharry Medical College, Nashville, TN 37208, USA; vsharma@mmc.edu (V.S.); okorolkova@mmc.edu (O.K.)

**Keywords:** endothelial dysfunction, periodontal disease, polybacterial infection, L-Sepiapterin, CDDO-Me, oxidative stress, apoptosis

## Abstract

**Highlights:**

**What are the main findings?**

**What is the implication of the main finding?**

**Abstract:**

Background: Polybacterial infections associated with periodontitis are increasingly linked to systemic vascular complications, yet the underlying endothelial mechanisms remain unclear. This study investigated how a consortium of red-complex bacteria (*Porphyromonas gingivalis*, *Tannerella forsythia*, *Treponema denticola*) and orange complex (*Fusobacterium nucleatum*) affects oxidative stress, inflammation, metabolism, and apoptosis in endothelial cells, and whether L-Sepiapterin [a tetrahydrobiopterin (BH4) precursor via salvage pathway] or bardoxolone methyl (CDDO-Me) [a potent nuclear factor erythroid 2-related factor 2 (Nrf2) activator)] could provide protection. Methods: Human umbilical vein endothelial cells (HUVECs) were infected for 12–72 h and treated with L-Sepiapterin or CDDO-Me. Nitric oxide (NO), BH4, and reactive oxygen species (ROS) levels were quantified, and mRNA expression of key genes regulating nitric oxide synthase activity, antioxidant defense, inflammation (TLR4/NF-κB, cytokines), metabolism (PI3K-AKT-PEA-15), and apoptosis (FAS–caspase pathway) was analyzed. Results: Infection markedly reduced NO and BH4, elevated ROS, activated TLR4/NF-κB and proinflammatory cytokines, disrupted PI3K/AKT signaling, and triggered endothelial apoptosis. Treatments with L-Sepiapterin and CDDO-Me restored NO bioavailability, reduced oxidative and inflammatory responses, normalized metabolic gene expression, and attenuated apoptosis, with CDDO-Me showing more promising effects. This study provides the mechanistic insight linking periodontal polybacterial infection to endothelial dysfunction and metabolic impairment such as diabetes, suggesting that redox-modulating strategies such as L-Sepiapterin and CDDO-Me may help prevent vascular damage associated with periodontal disease.

## 1. Introduction

Periodontal disease (PD) is a chronic inflammatory condition characterized by gingival inflammation and progressive destruction of the supporting periodontal tissues. It affects approximately 20–50% of the global population and about 42% of adults in the United States [1,2]. The “red complex” bacteria *Porphyromonas gingivalis* (*P. gingivalis*), *Tannerella forsythia* (*T. forsythia*), and *Treponema denticola* (*T. denticola*), together with *Fusobacterium nucleatum* (*F. nucleatum*) as a bridging species, play central roles in disease progression [3]. Beyond its local effects in the oral cavity, PD is increasingly recognized as a key contributor to systemic pathologies, including cardiovascular, metabolic, neurological disorders, and oral cancers [4,5]. The translocation of periodontal pathogens and their virulence factors, along with the systemic spread of inflammatory mediators from diseased periodontal tissues, can trigger endothelial dysfunction, oxidative stress, and chronic systemic inflammation [6]. Collectively, these pathophysiological mechanisms underscore PD as a critical risk factor contributing to the onset and progression of vascular and metabolic disorders.

One of the principal mechanisms linking PD to systemic diseases is vascular endothelial dysfunction, an early pathogenic event in cardiovascular and metabolic disorders. A healthy vascular endothelium is essential for maintaining cardiovascular homeostasis, regulating vascular tone, coagulation, permeability, and angiogenesis through signaling molecules such as nitric oxide (NO) [7,8]. Endothelial NO synthase (eNOS)-derived NO promotes vasodilation, anti-inflammatory signaling, and inhibition of thrombosis, whereas proinflammatory cytokines and bacterial components during infection induce inducible NOS (iNOS), resulting in excessive NO production, oxidative stress, and endothelial injury [9,10]. The redox-sensitive transcription factor Nrf2 regulates antioxidant defenses by regulating genes and its dysregulation exacerbates oxidative stress, inflammation, and endothelial dysfunction [11]. Chronic exposure to periodontal pathogens, particularly red-complex bacteria, along with their virulence factors such as enzymes, toxins, and various outer membrane proteins (capsule, fimbriae, lipopolysaccharide (LPS), gingipain, dentilisin, Fusobacterium adhesin A (FadA), etc.), triggers apoptotic signaling cascades and disrupts redox homeostasis, simultaneously activating nuclear factor kappa-B (NF-κB)-mediated proinflammatory pathways [12,13]. Furthermore, these pathogens induce systemic inflammation and oxidative stress, impairing both NO bioavailability and vascular homeostasis, leading to the progression of atherogenesis and hypertension [14]. Moreover, periodontal pathogens can impair endothelial insulin signaling by disrupting the PI3K–AKT pathway and modulating downstream effectors such as phosphoprotein enriched in astrocytes-15 (PEA-15). PEA-15 is a multifunctional regulator of cell proliferation, apoptosis, and glucose transport signaling. Altered PEA-15 expression influences insulin resistance, thereby linking PD-mediated metabolic diseases such as type 2 diabetes mellitus (T2DM) [15,16,17].

L-Sepiapterin, a natural precursor of tetrahydrobiopterin (BH4), protects endothelial cells by restoring eNOS coupling and preventing the shift from NO to superoxide production [18]. Through the BH4 salvage pathway and regeneration by dihydrofolate reductase (DHFR), L-Sepiapterin sustains sufficient BH4 for eNOS dimerization and NO synthesis, thereby reducing reactive oxygen species (ROS) formation and oxidative stress [19]. Similarly, CDDO-Me (bardoxolone methyl), a synthetic triterpenoid, exerts potent antioxidant and anti-inflammatory effects by activating the Nrf2–antioxidant response element (ARE) pathway via Kelch-like ECH-associated protein 1 (KEAP1) inhibition and suppressing NF-κB signaling through the inhibition of IκB kinase β (IKKβ) [20]. Together, these mechanisms preserve redox homeostasis, attenuate inflammation, and protect endothelial cells from infection-induced oxidative injury and apoptosis.

Although the association between PD and endothelial dysfunction is well documented [12], the specific molecular mechanisms underlying polybacterial infection-induced endothelial cell dysfunction remain underexplored. Our previous work showed that polybacterial infection markedly impairs NO synthesis and alters oxidative and inflammatory biomarkers. The findings highlighted the vascular risk posed by polybacterial periodontal infection compared to single-species exposure [10]. However, the potential of redox-regulating agents such as L-Sepiapterin and CDDO-Me to restore endothelial homeostasis under these conditions has not been studied. Therefore, this study addresses a critical knowledge gap by elucidating how these agents modulate the eNOS/BH4/NO axis, Nrf2–ARE pathway, and PI3K–AKT–IRS–GLUT signaling to counteract infection-induced oxidative stress, inflammation, and impaired metabolic signaling in endothelial cells. We hypothesize that polymicrobial infection disrupts endothelial function through reduced NO bioavailability, oxidative stress, inflammation, and impaired metabolic signaling, whereas treatment with either agent would restore endothelial integrity by re-establishing eNOS/BH4/NO coupling, activating Nrf2-mediated antioxidant defenses, and normalizing PI3K–AKT–IRS–GLUT signaling. In this study, we investigated the effects of polybacterial periodontal infection and evaluated the mechanistic potential of either L-Sepiapterin or CDDO-Me as targeted therapeutic interventions using human umbilical vein endothelial cells (HUVECs). The HUVECs were chosen as a model because they are a well-established and physiologically relevant system for studying vascular endothelial function. HUVECs exhibit key characteristics of systemic endothelial cells, including NO production, inflammatory cytokine responses, and regulation. They are widely used to investigate mechanisms of endothelial dysfunction, oxidative stress, and inflammation associated with cardiovascular and metabolic diseases [21,22].

## 2. Materials and Methods

### 2.1. Pre-Culture of Bacteria

In this study, red-complex periodontal pathogens, *P. gingivalis* (ATCCW83), *T. denticola* (ATCC 35404), and *T. forsythia* (ATCC 43037), along with F. nucleatum (ATCC 49256), were revived from glycerol stocks stored at −80 °C. A 50 µL aliquot of each stock was individually inoculated into five milliliters of Oral Treponeme Enrichment Broth (OTEB; Anaerobe Systems, Morgan Hill, CA, USA). For *T. forsythia*, the medium was supplemented with N-acetylmuramic acid (NAM, 1 mg/mL). Cultures were incubated in a BACTRON-300 Anaerobic Chamber (Sheldon Manufacturing, Cornelius, OR, USA) at 37 °C for 24–48 h. Bacterial density was estimated by measuring the optical density (OD) at 600 nm (OD_600_). This was based on established OD–CFU correlations for these species (*P. gingivalis* OD_600_ = 1.0 ≈ 1 × 10^9^ CFU/mL; *F. nucleatum* OD_600_ = 1.0 ≈ 5 × 10^8^ CFU/mL; *T. denticola* and *T. forsythia* OD_600_ = 1.0 ≈ 9 × 10^7^ CFU/mL) [23,24,25,26]. Cultures were then combined proportionally to prepare a polybacterial suspension with a final concentration of ~10^8^–10^9^ CFU/mL.

### 2.2. Pre-Culture Primary Human Umbilical Vein Endothelial Cells (HUVEC)

Human umbilical vein endothelial cells (HUVECs; PCS-100-010™, ATCC, Rockville, MD, USA) were maintained in vascular cell basal medium (VCBM) supplemented with an Endothelial Cell Growth Kit-VEGF and 10% fetal bovine serum (FBS), following the manufacturer’s instructions (ATCC, Rockville, MD, USA). Cells were incubated at 37 °C in a humidified atmosphere containing 5% CO_2_. Approximately 5000 cells were seeded onto 24 × 40 mm glass coverslips and cultured until reaching over 80% confluence, at which point they were used for HoxBan co-culture experiments. All experiments were performed using confluent cells between passages two and four. Unless otherwise specified, all reagents were procured from ATCC and Fisher Scientific.

### 2.3. Human Oxygen–Bacteria Anaerobic (HoxBan) Co-Culture

The Human Oxygen–Bacteria Anaerobic (HoxBan) co-culture method was performed as previously described, with minor modifications [10]. Briefly, *P. gingivalis, T. denticola, T. forsythia*, and *F. nucleatum* were precultured in their respective media for 24–48 h to obtain exponentially growing cells. Cultures were harvested and washed with sterile PBS, and the optical density (OD_600_) was measured to adjust the bacterial concentration to approximately 10^8^–10^9^ CFU/mL. Each bacterial suspension was then inoculated into 100 mL of freshly autoclaved and cooled (~40 °C) OTEB medium supplemented with NAM broth containing 1% agar and 0.5% thioglycolate. Subsequently, 30 mL of this inoculum was poured into sterile 150 mm Petri plates, allowed to solidify for 30 min, and incubated in a BACTRON-300 Anaerobic Chamber at 37 °C for 48 h. Human umbilical vein endothelial cells (HUVECs) grown on coverslips (~80% confluency) were inverted onto the solidified agar surface to allow direct exposure to the actively growing polybacterial consortium. The setup was overlaid with 20 mL of pre-warmed (37 °C) complete VCBM medium (without antibiotics). The final bacterial concentration on the agar surface was estimated to yield a multiplicity of infection (MOI) of approximately 100 for each bacterial species.

To evaluate bacterial viability in the HoxBan co-culture system, the culture supernatant and coverslips were carefully removed from the agar surface at the terminal time point (72 h). A small section of the agar surface was then scraped and transferred into 10 mL of OTEB medium supplemented with NAM, followed by anaerobic incubation at 37 °C for 24 h. The culture was subsequently subcultured on respective bacterial media to assess growth and confirm bacterial viability throughout the experiment. Additionally, a separate experiment was conducted to assess the survival of each bacterial species in VCBM under conditions mimicking the HoxBan culture for up to 72 h. Samples were subsequently subcultured onto their respective growth media, and bacterial growth was observed at all time points. For treatment, either L-Sepiapterin (100 µM; Cayman Chemical, Ann Arbor, MI, USA) or CDDO-Me (100 nM; Cell Signaling Technology, Danvers, MA, USA) was used, as determined by the cell viability assay described in Section 3.1. Both compounds were dissolved in dimethyl sulfoxide (DMSO) to prepare the desired concentrations and subsequently diluted in pre-warmed (37 °C) complete VCBM medium (without antibiotics) before application. For the negative control (uninfected), HUVECs cultured on coverslips were placed upside down on the agar surface without bacterial inoculum. After setting up the cultures, the co-culture plates were incubated in a humidified incubator at 37 °C and 5% CO_2_ for various time points ranging from 12 to 72 h. At the end of each time point, culture supernatant was collected and stored at −80°C for subsequent experiments. The experiment was conducted in three independent biological replicates, each with two technical replicates. Schematic overview of experimental design is depicted in Figure 1.

### 2.4. L-Sepiapterin and CDDO-Me Toxicity Assay

MTT [3-(4,5-Dimethylthiazol-2-yl)-2,5-Diphenyltetrazolium Bromide] assay was utilized to evaluate the toxicity of the L-Sepiapterin and CDDO-Me to the HUVECs. Cells were incubated with the complete vascular basal media prepared as described above (Section 2.2) as a control, with different concentrations of L-Sepiapterin (25 to 200 μM) or CDDO-Me (25–300 nM) for 6 to 72 h. At the end of each time point, 50 μL of 5 mg/mL MTT reagent was added to the cells. The cells were subsequently incubated for 4 h at 37 °C until formation of formazan crystals. The formazan crystals were quantified by dissolving them in 500 μL of DMSO (Sigma Aldrich, St. Louis, MO, USA), and the absorbance was read at 570 nm using a microplate reader (BioTek Synergy HT, Winooski, VA, USA). Cell viability was calculated by normalizing the absorbance of treated wells to that of control wells, providing baseline correction for each time point. The highest concentrations exhibiting more than 80% cell viability were subsequently selected for further experiments.

### 2.5. Nitric Oxide (NO) Quantification

HUVECs were pretreated with L-Sepiapterin (100 µM) or CDDO-Me (100 nM) for 3 h at 37 °C prior to infection. Following pretreatment, the cells on coverslips were subjected to polybacterial co-culture using the HoxBan method, as described in Section 2.3, for 12, 24, 48, and 72 h. At the end of each co-culture time point, NO levels in cell co-culture supernatants were quantified using the Total Nitric Oxide and Nitrate/Nitrite Assay Kit (R&D Systems, Minneapolis, MN, USA) according to the manufacturer’s protocol. The assay relies on enzymatic conversion of nitrate to nitrite by nitrate reductase, followed by colorimetric detection of nitrite via the Griess reaction. Supernatants were filtered using 10 kDa cut-off filters (Millipore Sigma, Cat# UFC501096, Visalia, CA, USA), diluted 1:5 with assay buffer, and incubated with NADH and nitrate reductase at 37 °C for 30 min, followed by sequential addition of Griess Reagents I and II. After a 10 min incubation at room temperature, absorbance was measured at 540 nm using a microplate reader (BioTek Synergy HT, Winooski, VA, USA). NO concentrations were calculated from a sodium nitrite standard curve using a four-parameter logistic (4-PL) fit in GraphPad Prism v9.

### 2.6. Quantification of Tetrahydrobiopterin (BH4) by ELISA

HUVECs were infected and treated as described above (Section 2.4). BH4 levels in cell co-culture supernatants for each time point were quantified using a commercial ELISA kit (Cat. no. NBP2-74983, Novus Biologicals LLC, Centennial, CA, USA) based on a competitive ELISA principle. Briefly, samples and standards (50 µL/well) were added to a 96-well microplate pre-coated with BH4, followed by the addition of 50 µL biotinylated detection antibody. After incubation at 37 °C for 45 min, wells were washed three times, and 100 µL of HRP conjugate was added and incubated for 30 min at 37 °C. Plates were washed five times, and 90 µL of TMB substrate solution was added to each well, followed by a 15 min incubation at 37 °C in the dark. The reaction was stopped with 50 µL of stop solution, and absorbance was measured at 450 nm using a microplate reader (BioTek Synergy HT, Winooski, VA, USA). Sample concentrations were calculated from a four-parameter logistic (4-PL) standard curve generated using GraphPad Prism v9.

### 2.7. Extracellular H_2_O_2_ Measurement (ROS)

Extracellular hydrogen peroxide (H_2_O_2_) levels in culture supernatants were determined using the ROS-Glo™ H_2_O_2_ Assay (Cat. no. G8820, Promega, Madison, WI, USA). HUVECs were infected and treated as described above (Section 2.4). At each time point, media from HoxBan co-cultures were collected and assayed according to the manufacturer’s instructions. In brief, 80 µL of culture media was transferred to a white opaque 96-well plate, mixed with 20 µL H_2_O_2_ substrate solution, and incubated for 60 min at room temperature. Subsequently, 100 µL of ROS-Glo™ detection solution was added and incubated for an additional 20 min. Luminescence was recorded using a microplate reader (BioTek Synergy HT, Winooski, VA, USA), and values were corrected by subtracting enzyme-independent background signals from enzyme-active reactions. All experiments were conducted in three biological replicates and two technical replicates.

### 2.8. RNA Isolation and Quantitative Real-Time Polymerase Chain Reaction (PCR) Analysis

HUVECs were infected and treated as described above (Section 2.4). At the end of the co-culture experiment, coverslips were removed from the agar surface, and adherent HUVECs were processed for RNA extraction using the TRIzol method as described previously [10]. The quality of RNA was determined by NanoDrop (Thermo Fisher Scientific, Waltham, MA, USA) spectrophotometry by evaluating A260/A280 and A260/A230 ratios, and only samples with acceptable purity values were used for cDNA synthesis. RNA was reverse-transcribed to cDNA using the iScript cDNA Synthesis Kit (Bio-Rad, Richmond, CA, USA). The primers used to detect various biomarkers are depicted in Table 1. All primers were synthesized in IDT Morrisville, NC, USA. Quantitative real-time polymerase chain reaction assay was performed using iQ SYBR Green Supermix (Bio-Rad, Richmond, CA, USA). The PCR cycle started at 50 °C for 2 min and 95 °C for 3 min, followed by 40 cycles of 95.0 °C for 15 s and 60 °C for 1 min. The specificity and reliability of qPCR reactions were verified through melt curve analysis. The number of the threshold cycle (Ct) for each target mRNA was normalized to the corresponding Ct of β-actin mRNA to obtain the ΔCt, and then 2^−ΔΔCt^ was calculated to obtain the relative mRNA fold change against the control. qPCR data were analyzed using Bio-Rad CFX Maestro Version 2. Three independent experiments were performed with at least two technical replicates. All experiments were performed in the Meharry Medical College Molecular Biology Core Laboratory.

### 2.9. Terminal Deoxynucleotidyltransferase-Mediated dUTP Nick-End Labeling (TUNEL) Assay

HUVECs were seeded in 35 mm dishes (Ibidi, Fitchburg, WI, USA) at a density of 1.5 × 10^5^ cells/dish in 1 mL of VCBM and incubated for 24 h. The medium was then removed, and cells were washed twice with 1× PBS. HUVECs were pretreated with L-Sepiapterin (100 µM) or CDDO-Me (100 nM) in antibiotic-free VCBM for 3 h, followed by infection with an actively growing polybacterial suspension (prepared in 1× PBS) at a multiplicity of infection (MOI) of 100 for each bacterial species. Cells were incubated at 37 °C in 5% CO_2_ for 24 h. Untreated infected cells served as positive controls, whereas uninfected cells served as negative controls. After 24 h of infection, the medium was discarded, and cells were washed three times with 1× PBS before proceeding with the TUNEL assay using the In Situ Cell Death Detection Kit, Fluorescein (Roche Diagnostics GmbH, Mannheim, Germany), following the manufacturer’s instructions. Briefly, cells were fixed in 4% formaldehyde (in PBS) for 15 min at room temperature, washed three times with PBS, and permeabilized with 0.1% Triton X-100 in 0.1% sodium citrate for 8 min at room temperature. Cells were then incubated in freshly prepared TUNEL reaction mixture for 60 min at 37 °C in the dark, washed three times with PBS, air-dried, and mounted with ProLong™ Diamond Antifade Mountant containing DAPI (Thermo Fisher Scientific, Eugene, OR, USA). Fluorescent images were captured using a Nikon AX Ti2 confocal microscope (Nikon, Melville, NY, USA) with a 20× objective lens, and analysis was performed using NIS-Elements software (v5.42.03, 64-bit). For this, a pipeline was created in the general analysis module within NIS-Elements. Thresholding in the 405 (DAPI) channel was used to identify nuclei, and then false positives were removed by filtering recognized objects by size. Then, nuclei exhibiting green fluorescence above the threshold set up in the 488 channel were counted as apoptotic. Apoptosis was quantified by counting TUNEL-positive cells from at least six randomly selected fields per sample across three independent biological replicates and expressed as the percentage of total cells counted.

### 2.10. Statistical Analysis

Statistical analysis was conducted using GraphPad Prism (GraphPad Software, version 9, Boston, MA, USA). Data normality was assessed using the Shapiro–Wilk test, and Q–Q plots were used for visual inspection of distribution. Depending on the comparison, either Student’s *t*-test or two-way ANOVA was applied, followed by Tukey’s multiple comparison post hoc test to determine group-wise significance. A *p*-value of less than 0.05 was considered statistically significant. * *p* < 0.05 compared with control (uninfected) cells, and # *p* < 0.05 compared with polybacterial infection. Data are presented as mean ± SEM of three independent biological replicates, each with at least two technical replicates, unless otherwise stated.

## 3. Results

### 3.1. Cell Viability with L-Sepiapterin and CDDO-Me in HUVECs

We first evaluated the cytotoxicity of the pharmacological agents L-Sepiapterin and CDDO-Me in HUVECs to establish safe working concentrations for subsequent experiments. To evaluate the potential cytotoxicity of L-Sepiapterin and CDDO-Me in HUVECs, cells were treated with increasing concentrations of each compound for 6–72 h, and viability was assessed using the MTT assay. As shown in Figure 2a, treatment with L-Sepiapterin up to 100 µM did not reduce cell viability below 90 ± 3.8% at any time point. However, at concentrations ≥100 µM, a statistically significant (*p* < 0.05) reduction in viability was observed at 24 and 72 h compared to the control, with 200 µM reducing viability to 77 ± 3.4% after 48 h, indicating dose-dependent cytotoxicity at higher doses. Similarly, Figure 2b shows that CDDO-Me maintained cell viability above 85 ± 2.5% at concentrations up to 100 nM during the 72 h incubation period. Although a transient significant reduction was observed at 24 h for 100 nM, no significant decline occurred at later time points. In contrast, concentrations above 200 nM resulted in a marked and inconsistent decrease in cell viability. Collectively, these findings indicate that L-Sepiapterin ≤ 100 µM and CDDO-Me ≤ 100 nM were well tolerated by HUVECs and were therefore selected as the working concentrations for subsequent experiments.

### 3.2. Effects of Polybacterial Infection and L-Sepiapterin or CDDO-Me Treatment on NO, BH4 Levels, and NOS Expression

After establishing a non-toxic concentration of L-Sepiapterin and CDDO-Me in HUVECs, we next examined how polybacterial infection affects endothelial nitric oxide production and NOS signaling, and whether the compounds could restore NO homeostasis.

The impact of polybacterial infection on endothelial NO synthesis was evaluated by quantifying NO and BH4 levels in co-culture supernatants and mRNA expression NOS signaling at 12–72 h. This approach allowed us to assess the biochemical and transcriptional impact of infection on eNOS/iNOS signaling. We further evaluated whether pharmacological interventions with L-Sepiapterin or CDDO-Me could restore NO production and normalize NOS-related gene expression. As shown in Figure 3a,b, polybacterial infection reduced both BH4 and NO production significantly (*p* < 0.05) at all time points compared with uninfected controls, indicating disruption of endothelial nitric oxide homeostasis. Treatment with L-Sepiapterin significantly restored BH4 levels (*p* < 0.05) at all measured time points and enhanced NO production, with a significant increase observed after 48 h compared with infected cells (*p* < 0.05). These results are consistent with the known role of L-Sepiapterin as a BH4 precursor via the salvage pathway, effectively supporting eNOS-mediated NO synthesis. Similarly, CDDO-Me also improved BH4 and NO availability, showing a significant increase in BH4 levels at early time points (12 and 24 h; *p* < 0.05). However, at later time points, the restorative effects of CDDO-Me were less pronounced compared with L-Sepiapterin.

At the transcriptional level as shown in Figure 3c,d, infection induced significant downregulation of eNOS mRNA and upregulation of iNOS mRNA (*p* < 0.05) as compared to control, reflecting a shift toward dysfunctional nitric oxide synthase signaling. L-Sepiapterin treatment significantly enhanced eNOS expression while attenuating iNOS induction (*p* < 0.05). Likewise, CDDO-Me also produced a corrective effect, markedly reducing iNOS mRNA expression at all time points and partially restoring eNOS expression at later time points. Collectively, these findings indicate that polybacterial infection impairs endothelial nitric oxide production through BH4 depletion and NOS uncoupling. On the other hand, L-Sepiapterin and CDDO-Me restored this phenotype. Similarly, the expression of DHFR (salvage pathway for BH4 synthesis) and GTP cyclohydrolase 1 (GCH-1; de novo BH4 synthesis) was analyzed (Figure 3e,f). DHFR mRNA levels decreased in a time-dependent manner, reaching a significant reduction (*p* < 0.05) at 48 h post-infection. Treatment with L-Sepiapterin and CDDO-Me restored DHFR expression, with CDDO-Me showing a significant (*p* < 0.05) effect at 24 h and L-Sepiapterin at 48 h. In contrast, GCH-1 mRNA exhibited a slight increase following infection, which gradually returned to near-basal levels; however, these changes were not statistically significant (*p* > 0.05). Together, these findings indicate that polybacterial infection impairs NO synthesis and disrupts normal NOS signaling in endothelial cells. L-Sepiapterin provided the strongest and most sustained restoration of BH4 and eNOS activity, whereas CDDO-Me was effective primarily at early time points; both agents consistently reduced iNOS induction, highlighting their relative efficacy in correcting infection-induced endothelial dysfunction.

### 3.3. Polybacterial Infection-Induced Oxidative Stress and Modulation by Treatments

Given the critical role of redox balance in endothelial function, we analyzed Nrf2-mediated antioxidant responses and intracellular ROS levels following infection and treatment. To assess the redox regulatory response to polybacterial infection, we first analyzed Nrf2 expression and downstream antioxidant activity in HUVECs. This approach allowed us to evaluate how infection disrupts the cellular antioxidant defense system and whether L-Sepiapterin or CDDO-Me can restore redox balance. As shown in Figure 4a, infection suppressed Nrf2 expression at all time points (12–72 h; *p* < 0.05 vs. control). Treatment with either L-Sepiapterin or CDDO-Me restored Nrf2 expression, although the effect of L-Sepiapterin was modest within the first 24 h. From 48 h onward, both compounds significantly enhanced Nrf2 expression levels (*p* < 0.05), indicating activation of the cellular antioxidant defense pathway.

To further assess oxidative imbalance, intracellular ROS levels were quantified (Figure 4b). Polymicrobial infection markedly increased ROS generation at all time points examined (12–72 h; *p* < 0.01 vs. control). Treatments attenuated ROS accumulation, with CDDO-Me demonstrating an earlier and stronger antioxidant response. ROS levels were significantly reduced as early as 24 h in CDDO-Me-treated cells and at 48 h and 72 h following L-Sepiapterin treatment (*p* < 0.05). Subsequently, Nrf2 downstream antioxidant enzymes, GCLC and SOD-1, were evaluated to confirm transcriptional activation of the Nrf2 pathway. Both GCLC and SOD-1 mRNA levels declined within 12 h of infection, reaching significantly reduced levels at 48 h (0.5-fold) and 72 h (0.49-fold), respectively, as compared to the control (*p* < 0.05). Treatment with either compound restored their expression, with CDDO-Me producing a more pronounced and sustained effect, significantly upregulating mRNA levels from 12 h through 72 h (*p* < 0.05). L-Sepiapterin also enhanced GCLC and SOD-1 expression at all time points, achieving statistical significance after 48 h of treatment (Figure 4c,d).

Overall, these findings suggest that polybacterial infection induces robust oxidative stress and suppresses Nrf2-mediated antioxidant defenses. Treatment with L-Sepiapterin and CDDO-Me mitigated these effects, with CDDO-Me providing a faster and more sustained activation of the Nrf2–GCLC/SOD-1 axis, highlighting its superior efficacy in restoring redox homeostasis.

### 3.4. Polybacterial Infection Activates TLR4/NF-κB-Mediated Inflammatory Signaling, Attenuated by L-Sepiapterin and CDDO-Me

Following the assessment of oxidative stress, we investigated the inflammatory response induced by polybacterial infection in HUVECs. Key upstream receptors (TLR4, C5aR1) and downstream cytokines were analyzed to determine how infection activates TLR4–NF-κB- and GPCR-mediated pathways, and to evaluate the modulatory effects of L-Sepiapterin and CDDO-Me. As shown in Figure 5a,b, both TLR4 and NF-κB mRNA levels were significantly (*p* < 0.05) elevated in a time-dependent manner following infection, with NF-κB showing a peak 13.4-fold increase at 48 h and TLR4 reaching a maximum 7.4-fold induction at 72 h compared with uninfected controls. Treatment with either L-Sepiapterin or CDDO-Me effectively attenuated this upregulation, with CDDO-Me producing a more pronounced and earlier inhibitory effect. NF-κB expression was significantly reduced as early as 12 h and remained suppressed throughout 72 h (*p* < 0.05), whereas TLR4 downregulation became significant after 48 h with the treatments. In addition, the complement receptor C5aR1, a prototype G protein-coupled receptor (GPCR) involved in inflammatory signaling, was significantly upregulated during infection (Figure 5c). C5aR1 expression was significantly upregulated at all time points (*p* < 0.05), peaking with a 2.6-fold increase at 48 h compared to controls, indicating complement activation and GPCR-mediated response to polybacterial infection.

Consistent with these findings, infection induced robust upregulation of proinflammatory cytokines TNF-α, IL-1α, IL-6, and IL-8 (Figure 5d–g). Expression levels peaked at 48 h for TNF-α (23-fold), 24 h for IL-1α (61-fold), IL-6 (31-fold), and IL-8 (15.5-fold) before declining slightly at 72 h yet remaining elevated relative to controls (*p* < 0.05). Treatments significantly mitigated this cytokine response, with CDDO-Me showing greater efficacy, likely through suppression of NF-κB-dependent transcription.

Overall, polybacterial infection strongly activates TLR4–NF-κB and C5aR1–GPCR signaling, driving a robust proinflammatory cytokine response. L-Sepiapterin and CDDO-Me effectively attenuated these pathways, with CDDO-Me demonstrating faster and more pronounced anti-inflammatory effects, highlighting their potential in restoring endothelial inflammatory balance.

### 3.5. Polybacterial Infection Disrupts the PI3K–AKT–Insulin Signaling Axis, Restored by L-Sepiapterin and CDDO-Me

Since inflammation and oxidative stress can disrupt endothelial metabolism, we next assessed the impact of polybacterial infection on metabolic signaling, particularly insulin signaling and glucose transport in HUVECs. mRNA expression of key components of the PI3K–AKT–IRS–GLUT1–PEA-15 pathway was quantified to evaluate how infection alters metabolic signaling and regulatory feedback, and to determine whether L-Sepiapterin or CDDO-Me can restore metabolic homeostasis. As shown in Figure 6a,b, infection induced an early activation of the PI3K–AKT pathway, with both PI3K and AKT transcripts significantly upregulated at 12 h and 24 h (*p* < 0.05). However, prolonged infection led to a progressive decline, with AKT levels falling below basal values (0.6-fold) by 72 h, while PI3K expression returned to near baseline. Treatment with L-Sepiapterin or CDDO-Me effectively normalized these alterations and significantly upregulated AKT expression at 72 h (*p* < 0.05), indicating restoration of metabolic signaling activity.

GLUT1 expression (Figure 6c) decreased markedly following infection from 12 h to 72 h, consistent with impaired glucose transport. Although treatments elevated GLUT1 levels at all time points, the changes were not statistically significant (*p* > 0.05). Similarly, IRS1 and IRS2 showed sustained downregulation throughout 12–72 h post-infection (Figure 6d,e). IRS1 expression significantly declined after 24 h and remained low until 72 h (*p* < 0.05) but was restored by treatments. IRS2 followed a similar trend with modest recovery after treatment, though not statistically significant (*p* > 0.05). In contrast, PEA-15, a known inhibitor of AKT phosphorylation and mediator of insulin resistance, was significantly (*p* < 0.05) upregulated at all time points, peaking at 12 h (3.1-fold; *p* < 0.05) and remaining elevated through 72 h (Figure 6f). Treatment with L-Sepiapterin and CDDO-Me markedly reduced PEA-15 expression, with CDDO-Me exhibiting a stronger suppressive effect.

Collectively, these findings indicate that polybacterial infection disrupts the PI3K–AKT–PEA-15–IRS–GLUT1 signaling axis, contributing to endothelial insulin resistance and impaired glucose handling. L-Sepiapterin and CDDO-Me effectively restored this pathway, with CDDO-Me showing greater efficacy in suppressing PEA-15 and reactivating AKT, highlighting their potential to counteract infection-induced metabolic dysfunction.

### 3.6. Polybacterial Infection Induces Endothelial Apoptosis via the Fas–Caspase Pathway, Attenuated by L-Sepiapterin and CDDO-Me

To investigate whether polybacterial infection-induced oxidative, inflammatory, and metabolic disturbances lead to endothelial cell death, we analyzed key components of the Fas–caspase signaling cascade including upstream receptors, anti-apoptotic regulators, and downstream caspases and assessed DNA fragmentation to confirm apoptosis and evaluate the protective effects of L-Sepiapterin and CDDO-Me. Results showed that both FAS and FASL mRNA levels were markedly upregulated following infection, with expression significantly elevated from 24 h through 72 h (*p* < 0.05). Peak expression was observed at 24 h (FAS: 3.7-fold; FASL: 9.98-fold), followed by a gradual decline over time, though levels remained significantly higher than controls (*p* < 0.05), as shown in Figure 7a,b. Treatment with either L-Sepiapterin or CDDO-Me significantly reduced the infection-induced upregulation (*p* < 0.05), with CDDO-Me exhibiting a more pronounced suppressive effect.

The expression of c-FLIP, an anti-apoptotic regulator that inhibits caspase-8 activation, was consistently decreased in infected cells at all time points, reaching significance at 24 h as compared to control (*p* < 0.05) (Figure 7c). Treatments restored c-FLIP expression, with L-Sepiapterin showing a delayed but significant recovery at 72 h, while CDDO-Me partially restored expression earlier in the time course. Downstream apoptotic mediators caspase-8 and caspase-3 were also upregulated upon infection (Figure 7d,e). Caspase-8 expression was significantly elevated as early as 24 h (*p* < 0.05), indicating early activation of the extrinsic apoptotic pathway, whereas caspase-3 increased significantly at 48 h and remained elevated through 72 h. Treatment with CDDO-Me effectively suppressed caspase-3 activation beginning at 48 h (*p* < 0.05), while L-Sepiapterin reduced expression levels without reaching statistical significance. For caspase-8, treatments consistently reduced expression to near-basal levels from 24 h to 72 h (*p* < 0.05).

To verify these findings, apoptotic cell death was further evaluated using the TUNEL assay at 24 h after infection and treatments. (Figure 8a,b). Polybacterial infection led to a significant rise in TUNEL-positive nuclei, reflecting increased DNA fragmentation and apoptotic cell death (*p* < 0.05 vs. control). Treatment with L-Sepiapterin or CDDO-Me markedly reduced TUNEL-positive cells, demonstrating a significant anti-apoptotic and cytoprotective effect compared to infected cells (*p* < 0.05).

Collectively, these results indicate that polybacterial infection triggers robust endothelial apoptosis through Fas–caspase-8/3 activation and suppression of c-FLIP. Both L-Sepiapterin and CDDO-Me mitigated apoptosis, with CDDO-Me demonstrating a faster and stronger protective effect, highlighting their ability to preserve endothelial survival under infection-induced stress.

## 4. Discussion

This novel and innovative study, for the first time, elucidates the intricate molecular mechanisms by which periodontal polybacterial infection, consisting of *P. gingivalis*, *T. forsythia*, *T. denticola*, and *F. nucleatum*, induces endothelial dysfunction through oxidative stress, inflammatory activation, metabolic derangement, and apoptosis using HUVECs as a model. These four keystone pathogens, often associated with periodontitis, possess virulence factors such as gingipains, lipopolysaccharides, and hemagglutinins that enable them to invade systemic circulation, thereby influencing vascular integrity and promoting systemic inflammation [12,27,28]. This study provides a mechanistic link between periodontal infection and systemic vascular and metabolic dysfunction. We show that polybacterial infection simultaneously disrupts nitric oxide metabolism, elevates oxidative stress, activates inflammatory pathways, impairs insulin signaling, and triggers apoptosis in endothelial cells. Importantly, treatment with L-Sepiapterin, a BH4 precursor that restores eNOS coupling, and CDDO-Me, an Nrf2 activator that enhances antioxidant defense and metabolic balance, complementarily restores endothelial homeostasis, highlighting a promising therapeutic strategy to mitigate vascular damage induced by periodontal pathogens.

### 4.1. Nitric Oxide and BH4 Metabolism

Endothelial nitric oxide synthase (eNOS)-derived nitric oxide (NO) plays a pivotal role in maintaining vascular tone, redox balance, and overall endothelial homeostasis [29]. The current findings demonstrate that polybacterial infection strongly disrupts endothelial NO metabolism by reducing both BH4 and NO levels. This coordinated decline indicates eNOS uncoupling, a pathological shift wherein eNOS produces superoxide rather than NO, leading to endothelial oxidative stress and dysfunction. These observations are in agreement with previous reports showing that periodontal pathogens or their virulence factors, such as lipopolysaccharides (LPS) and gingipains, develop atherosclerosis as well as impair eNOS function by altering BH4 bioavailability and enhancing reactive oxygen species (ROS) generation [30,31,32,33]. This mechanistic alteration aligned with findings from previous studies showing that *P. gingivalis* lipopolysaccharide (LPS), outer membrane vesicles (OMVs), and other virulence factors disrupt endothelial NO signaling through inhibition of eNOS phosphorylation and impaired BH4 recycling [34,35], further supporting the notion that periodontal pathogens compromise endothelial NO bioavailability via cofactor depletion and redox imbalance [36]. Furthermore, the suppression of eNOS and simultaneous induction of inducible nitric oxide synthase (iNOS) further reinforce the concept of NOS dysregulation, where loss of eNOS-derived NO and overproduction of iNOS-derived NO jointly contribute to nitrosative stress and endothelial injury [37].

Therapeutic modulation with L-Sepiapterin significantly restored BH4 and NO production, reflecting its role as a direct precursor in the BH4 salvage pathway and its capacity to recouple eNOS activity. These results align with prior evidence that exogenous BH4 or its precursors prevent endothelial dysfunction by stabilizing eNOS dimer formation and reducing superoxide damage [18]. In contrast, CDDO-Me demonstrated an earlier restoration of BH4 levels, likely mediated through activation of the Nrf2 pathway, which regulates redox enzymes, including DHFR, and promotes antioxidant defenses [20]. Interestingly, the infection-induced downregulation of DHFR with minimal change in GCH1 expression suggests that endothelial BH4 deficiency primarily results from the salvage pathway (increased oxidized biopterins over BH4 levels) rather than decreased de novo synthesis. Restoration of DHFR expression by L-Sepiapterin and CDDO-Me highlights their role in metabolic and transcriptional regulation, respectively, to restore BH4 homeostasis. Together, these results highlight the complex role of nitric oxide metabolism in infection-induced endothelial dysfunction and suggest that combined interventions targeting BH4 restoration and redox regulation may represent a promising therapeutic strategy.

### 4.2. Nrf2-Mediated Redox Regulation

The results demonstrated that polybacterial infection impaired endothelial redox homeostasis by markedly suppressing Nrf2 expression and its downstream antioxidant genes, with a significant increase in intracellular ROS levels, indicative of a collapse in the Nrf2–ARE signaling axis. Such suppression of Nrf2 has been previously observed in vascular endothelial models exposed to *P. gingivalis* and related periodontal pathogens, which impair antioxidant defense and NO signaling, contributing to endothelial dysfunction and atherogenesis [36,38]. Treatment with L-Sepiapterin and CDDO-Me significantly restored Nrf2 expression and upregulated GCLC and SOD1, leading to attenuation of ROS accumulation. The protective effects of L-Sepiapterin are consistent with its ability to promote BH4-dependent eNOS coupling, thereby indirectly stabilizing redox balance, whereas CDDO-Me acts more directly by activating Nrf2 [39,40]. Our findings demonstrate that polybacterial infection disrupts endothelial redox homeostasis through coordinated suppression of Nrf2 and depletion of BH4, resulting in eNOS uncoupling and excessive ROS generation. This reciprocal imbalance between NO and ROS amplifies oxidative stress, further inhibiting NO bioavailability and exacerbating endothelial dysfunction. Pharmacological restoration of BH4 by L-Sepiapterin or activation of Nrf2 by CDDO-Me effectively interrupts this cycle, reducing ROS accumulation, supporting NO synthesis, and preserving endothelial redox balance. These results highlight the interdependence of NO and ROS pathways in vascular homeostasis and underscore the therapeutic potential of redox-modulating strategies to mitigate infection-induced endothelial damage.

### 4.3. Inflammatory and Complement Pathways

Polybacterial infection strongly activates the TLR4/NF-κB pathway in endothelial cells, leading to pronounced upregulation of proinflammatory cytokines (*TNF-α*, *IL-1α*, *IL-6*, *IL-8*), consistent with previous reports that *P. gingivalis* and other periodontal pathogens act through TLR4 to trigger NF-κB-dependent endothelial inflammation [12,27]. The early and sustained NF-κB activation observed here underscores its central role in linking microbial recognition to cytokine-mediated vascular dysfunction. In addition to TLR4–NF-κB activation, polybacterial infection markedly induced the complement receptor C5aR1, a prototypical G protein-coupled receptor (GPCR) that plays a central role in regulating inflammatory and immune responses. Activation of C5aR1 by its ligand C5a amplifies inflammation through crosstalk with TLR signaling, leading to enhanced NF-κB activation, cytokine release, and endothelial dysfunction [41,42]. Several studies have demonstrated that periodontal pathogens such as *P. gingivalis* can disrupt the complement–TLR axis by simultaneously engaging TLR4 and C5aR1, thereby promoting a dysregulated inflammatory response and impairing host defense mechanisms [42,43]. Notably, TLR2 (CD282) is another pattern recognition receptor that detects bacterial lipoproteins and components from *P. gingivalis* and *T. denticola* [44]. However, in our study, polybacterial infection did not induce noticeable changes in TLR2 mRNA expression, suggesting that the endothelial response was primarily mediated through the TLR4–C5aR1–NF-κB signaling axis rather than TLR2-dependent pathways. Treatment with L-Sepiapterin and CDDO-Me effectively suppressed this inflammatory response, with CDDO-Me showing a more potent and rapid effect. This aligns with prior studies demonstrating that CDDO-Me inhibits IKKβ-mediated NF-κB activation and reduces oxidative stress via Nrf2, thereby attenuating cytokine transcription [20,45]. L-Sepiapterin’s anti-inflammatory effect appears more indirect, likely mediated through restoration of eNOS-derived NO and redox balance, which can dampen NF-κB activation. Collectively, these results reinforce the mechanistic link between periodontal pathogens and endothelial inflammation and highlight the potential of redox-modulating agents to mitigate TLR4/NF-κB-driven vascular injury.

### 4.4. Metabolic and Apoptotic Dysregulation

Endothelial insulin signaling, mediated through the PI3K–AKT pathway, regulates glucose uptake, nitric oxide synthesis, and vascular homeostasis [46]. Our data revealed that infection transiently elevated *PI3K* and *AKT* at early stages (12–24 h), followed by a marked reduction at later time points, along with decreased *IRS1*, *IRS2*, and *GLUT1* expression. Such biphasic alteration suggests an initial adaptive activation followed by insulin resistance, potentially driven by periodontal pathogen-mediated inflammatory cytokines and oxidative stress. Notably, the upregulation of *PEA-15*, a known inhibitor of AKT phosphorylation, further confirms disruption of insulin signaling, as previously observed in diabetic and inflammatory endothelial injury models [16]. Treatments restored *AKT*, *IRS1*, and *GLUT1* expression, with CDDO-Me showing superior effects, possibly due to Nrf2-mediated modulation of metabolic gene networks [47]. L-Sepiapterin improves NO bioavailability, which could then enhance insulin sensitivity and glucose uptake via a different, more complex pathway, likely involving the correct insulin-responsive glucose transporter, GLUT1/4 [48,49]. However, in this study, GLUT4 mRNA expression was undetectable, suggesting that endothelial glucose transport under these conditions is primarily GLUT1-dependent, consistent with previous reports that GLUT4 expression is minimal or absent in endothelial cells [50]. Collectively, these data underscore the intricate interplay between periodontal pathogen-induced inflammation, oxidative stress, and metabolic dysregulation in infection-driven endothelial dysfunction.

Apoptosis is a major cause of endothelial injury and vascular instability [51]. The present study demonstrated that polybacterial infection upregulated *FAS* and *FASL* mRNA levels significantly (*p* < 0.05), accompanied by increased *caspase-8* and *caspase-3*, while *c-FLIP*, an apoptosis inhibitor, was downregulated. These alterations collectively indicate activation of the extrinsic Fas–caspase apoptotic pathway [52,53]. The TUNEL assay further confirmed extensive nuclear fragmentation, validating infection-induced apoptosis at the cellular level. Such apoptotic endothelial loss contributes to vascular leakage, inflammation, and impaired repair capacity [54]. L-Sepiapterin and CDDO-Me treatments markedly reduced Fas–caspase activation and apoptotic nuclei, with CDDO-Me showing more pronounced protection. Our result showed that polybacterial infection disrupts endothelial redox balance, suppressing Nrf2 and increasing ROS, which coincides with the upregulation of caspase-3, indicating that impaired antioxidant defenses drive the activation of the extrinsic apoptotic pathway. Mechanistically, Nrf2 likely mitigates apoptosis by upregulating anti-apoptotic genes such as *Bcl-2* and suppressing caspase cleavage [55], while eNOS activity further supports cell survival through anti-apoptotic signaling [56]. Treatment with CDDO-Me or BH4 supplementation via L-Sepiapterin restores Nrf2 activity reduces ROS accumulation and attenuates caspase-3 expression, demonstrating that modulation of redox and nitric oxide pathways can effectively counteract infection-induced endothelial apoptosis. Together, these findings highlight the integrated regulation of oxidative stress and apoptosis as a key mechanism by which these compounds preserve endothelial integrity during microbial insult.

### 4.5. Integrated Pathogenic Model and Therapeutic Implications

The findings of this study underscore that endothelial injury results from an interconnected network of oxidative, inflammatory, and metabolic disturbances rather than isolated mechanisms. The results provide important translational insights linking infection-induced endothelial dysfunction to clinically relevant vascular and metabolic disorders. Chronic exposure to periodontal pathogens can trigger endothelial oxidative stress, inflammation, and nitric oxide imbalance processes that contribute directly to atherosclerotic plaque formation, vascular stiffening, and insulin resistance [5]. The observed impairment in the PI3K–AKT–IRS signaling pathway and reduced GLUT1 expression parallel the mechanisms underlying endothelial insulin resistance and altered glucose uptake seen in diabetic vasculopathy [57,58]. Similarly, activation of the TLR4–NF-κB axis and Fas–caspase apoptotic signaling aligns with molecular events driving vascular inflammation and endothelial cell loss during atherogenesis [59,60]. Importantly, the restorative effects of L-Sepiapterin and CDDO-Me highlight a potential translational strategy to counteract these systemic effects. L-Sepiapterin restores eNOS coupling and nitric oxide bioavailability, while CDDO-Me activates Nrf2, enhancing antioxidant defense and metabolic regulation. These complementary mechanisms may not only preserve endothelial integrity but also attenuate the downstream risk of cardiovascular and metabolic complications associated with periodontitis. Thus, targeting endothelial redox and metabolic homeostasis offers a promising therapeutic avenue to mitigate the systemic vascular consequences of chronic oral infections. These integrated mechanisms and their broader clinical implications are summarized in Figure 9.

### 4.6. Limitations

This study has some limitations. Protein levels of all biomarkers were not measured, and RNA integrity (RIN) was not assessed, although RNA purity was confirmed using NanoDrop (A260/A280 and A260/A230 ratios) and melt curve analysis for nonspecific amplifications. In the HoxBan co-culture system, the precise MOI at the cell–agar interface cannot be determined, as bacterial contact is limited to the agar–coverslip interface, and some bacteria may remain in the medium without interacting. The estimated MOI should therefore be interpreted as approximate. Nonetheless, the HoxBan model provides a physiologically relevant polymicrobial environment, allowing host cells to be exposed to a complex bacterial community and its metabolites, which is the primary focus of this study.

## 5. Conclusions

This study elucidates that polybacterial infection induces endothelial dysfunction through interconnected oxidative, inflammatory, metabolic, and apoptotic pathways. Infection significantly disrupted eNOS coupling, depleted NO and BH_4_, elevated ROS, suppressed Nrf2 signaling, activated TLR4/NF-κB, impaired PI3K/AKT signaling, and triggered Fas–caspase-mediated apoptosis. L-Sepiapterin markedly restored NO levels and eNOS coupling, while CDDO-Me robustly enhanced Nrf2 activation, suppressed inflammation, and reduced apoptosis, collectively reversing endothelial injury.

By integrating these pathways, the study links periodontal pathogens to cardiovascular dysfunction, supporting the concept of periodontal-vascular crosstalk. Targeting dual redox and metabolic pathways represents a promising adjunctive therapeutic strategy. Future in vivo studies and clinical validation are warranted to translate these findings into interventions that mitigate cardiovascular risk in patients with chronic periodontitis.

## Figures and Tables

**Figure 1 cells-14-01777-f001:**
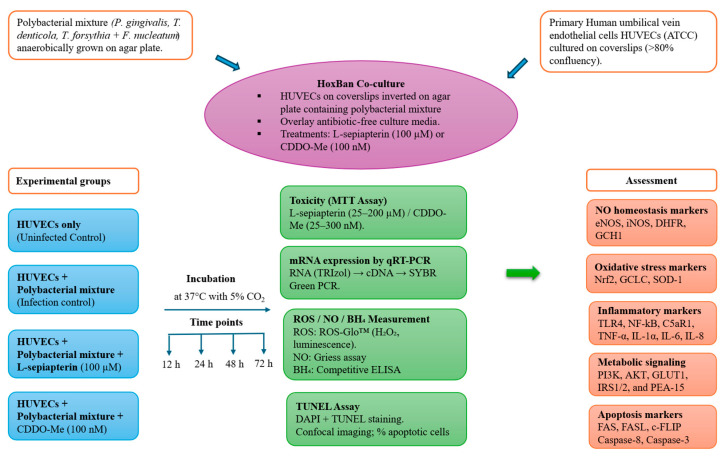
Schematic overview of experimental design. HUVECs were assigned to four experimental groups as uninfected control, polybacterial infection, infection + L-Sepiapterin (100 µM), and infection + CDDO-Me (100 nM). Treatments were administered 3 h prior to infection, followed by incubation for up to 72 h. Samples were collected at 12, 24, 48, and 72 h for downstream analysis. All experiments were conducted in biological triplicates with technical duplicates.

**Figure 2 cells-14-01777-f002:**
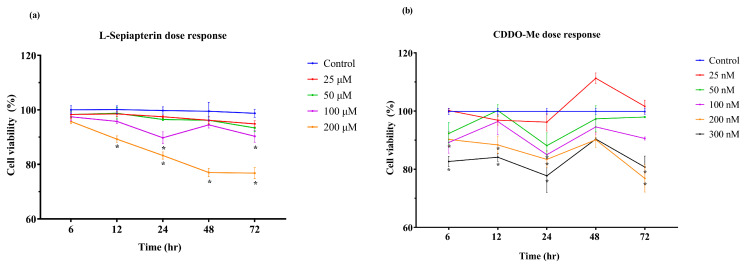
Dose-dependent effects of L-Sepiapterin and CDDO-Me on HUVEC viability. HUVECs were treated with increasing concentrations of (**a**) L-Sepiapterin (25–200 µM) or (**b**) CDDO-Me (25–300 nM) for 6–72 h. L-Sepiapterin maintained >90 ± 3.8% viability up to 100 µM, while CDDO-Me preserved >85 ± 2.5% viability up to 100 nM throughout the incubation period. Cell viability was measured by MTT assay and expressed as a percentage of control at each time point. Data represent mean ± SEM of three independent biological replicates and two technical replicates. * *p* < 0.05. Key: CDDO-Me = bardoxolone methyl.

**Figure 3 cells-14-01777-f003:**
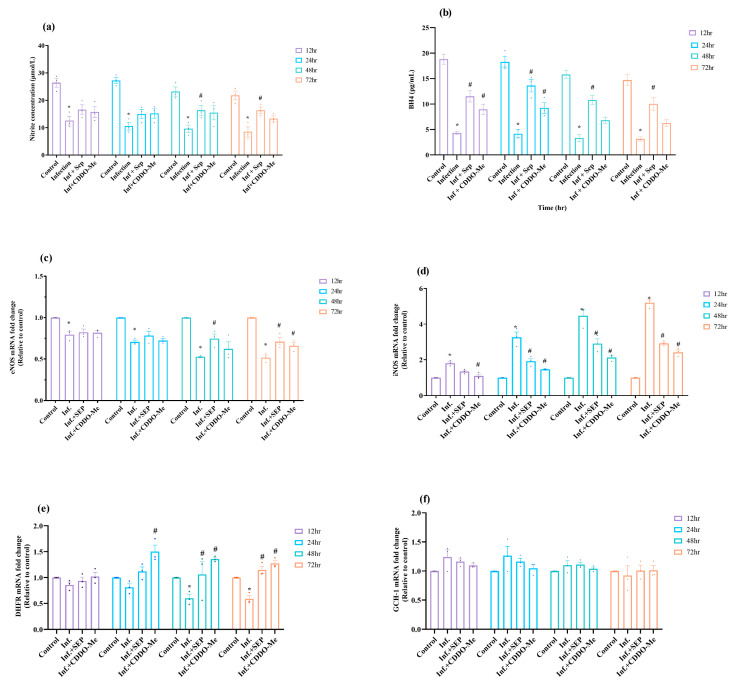
Polybacterial infection-induced endothelial dysfunction and its restoration by L-Sepiapterin and CDDO-Me. Human umbilical vein endothelial cells (HUVECs) were infected with a polymicrobial mixture of *P. gingivalis, T. denticola, T. forsythia*, and *F. nucleatum*, followed by treatment with L-Sepiapterin (100 µM) or CDDO-Me (100 nM) for 12–72 h). (**a**) BH4 and (**b**) NO levels in cell culture supernatants were quantified by ELISA. (**c**–**f**) mRNA expression of eNOS, iNOS, DHFR, and GCH-1 was assessed by quantitative real-time PCR, normalized to β-actin, and expressed as 2^−ΔΔCt^ relative to uninfected controls. Statistical significance was determined using two-way ANOVA followed by Tukey’s multiple comparisons. * *p* < 0.05 vs. control; # *p* < 0.05 vs. polybacterial infection. Data are presented as mean ± SEM of three independent biological replicates and two technical replicates. Abbreviations: Inf = infection; SEP = L-Sepiapterin; CDDO-Me = bardoxolone methyl; BH4 = tetrahydrobiopterin; eNOS = endothelial nitric oxide synthase; iNOS = inducible nitric oxide synthase; DHFR = dihydrofolate reductase; GCH-1 = GTP cyclohydrolase 1.

**Figure 4 cells-14-01777-f004:**
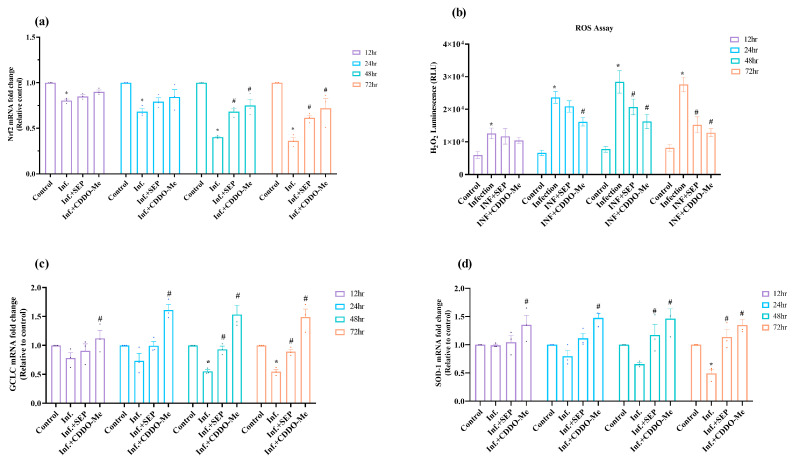
Polybacterial infection-induced oxidative stress and its modulation by L-Sepiapterin and CDDO-Me in HUVECs. HUVECs were infected and treated as above (Section 2.4). (**a**) Nrf2 mRNA expression; (**b**) ROS generation in culture supernatants; (**c**,**d**) GCLC and SOD-1 mRNA expression. mRNA levels were quantified by qRT-PCR, normalized to β-actin, and expressed as 2^−ΔΔCt^ relative to uninfected controls. ROS release was measured using the ROS-Glo™ H_2_O_2_ luminescent assay. Statistical significance was determined using two-way ANOVA followed by Tukey’s multiple comparisons. * *p* < 0.05 vs. control; # *p* < 0.05 vs. polybacterial infection. Data are presented as mean ± SEM of three independent biological replicates and two technical replicates. Abbreviations: Inf. = infection with polybacteria; SEP = L-Sepiapterin; CDDO-Me = bardoxolone methyl; ROS = reactive oxygen species; Nrf2 = nuclear factor erythroid-2 related factor 2; GCLC = glutamate–cysteine ligase catalytic subunit; SOD-1 = superoxide dismutase 1.

**Figure 5 cells-14-01777-f005:**
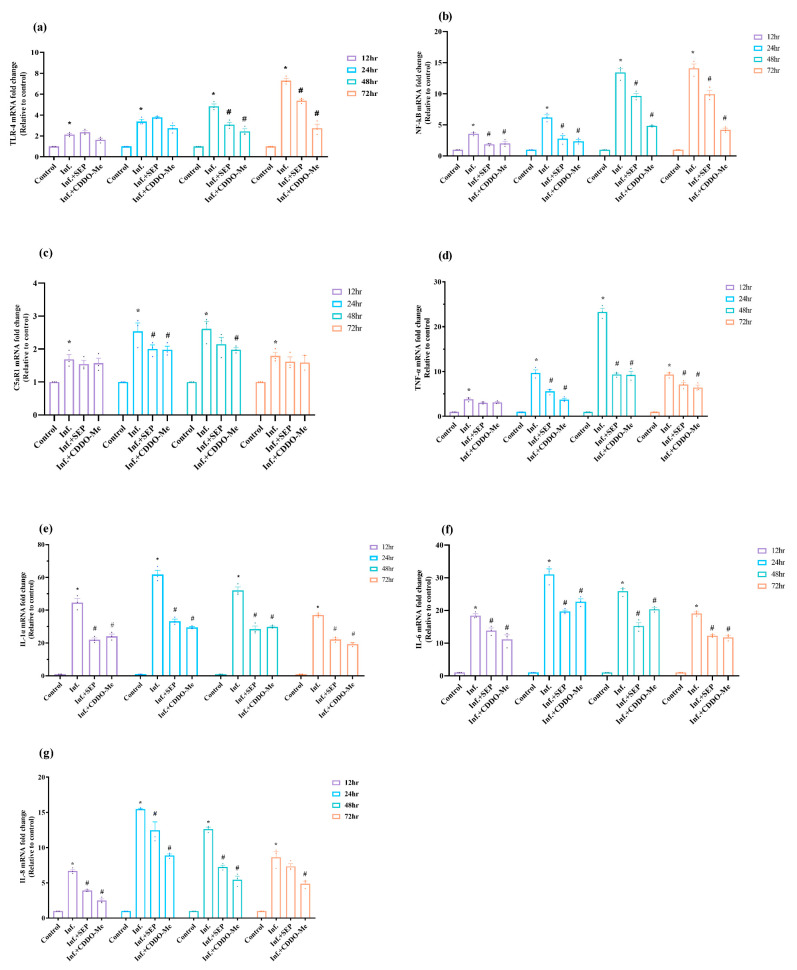
Polybacterial infection activates TLR4/NF-κB-mediated inflammatory signaling in HUVECs, attenuated by L-Sepiapterin and CDDO-Me. HUVECs were infected and treated as above (Section 2.4). (**a**–**c**) mRNA expression of TLR4, NF-κB, and C5aR1. (**d**–**g**) mRNA expression of proinflammatory cytokines TNF-α, IL-1α, IL-6, and IL-8. mRNA levels were quantified by qRT-PCR, normalized to β-actin, and expressed as 2^−ΔΔCt^ relative to uninfected controls. Statistical significance was determined using two-way ANOVA followed by Tukey’s multiple comparisons. * *p* < 0.05 vs. control; # *p* < 0.05 vs. polybacterial infection. Data are presented as mean ± SEM of three independent biological replicates and two technical replicates. Abbreviations: Inf. = infection with polybacteria; SEP = L-Sepiapterin; CDDO-Me = bardoxolone methyl; NF-κB = nuclear factor kappa-light-chain-enhancer of activated B cells; TLR4 = Toll-like receptor 4; TNF-α = tumor necrosis factor alpha; IL = interleukin; C5aR1 = complement component 5a receptor 1.

**Figure 6 cells-14-01777-f006:**
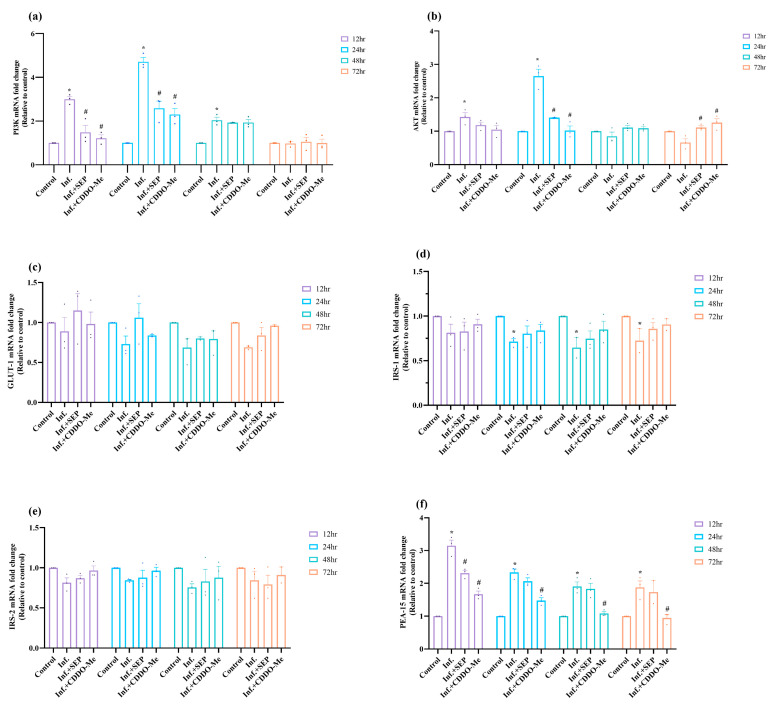
Polybacterial infection disrupts PI3K–AKT–insulin signaling in HUVECs, restored by L-Sepiapterin and CDDO-Me. HUVECs were infected and treated as above (Section 2.4). (**a**,**b**) mRNA expression of PI3K and AKT. (**c**–**e**) mRNA expression of GLUT1, IRS1, and IRS2, respectively. (**f**) Level of mRNA expression of PEA-15. mRNA levels were quantified by qRT-PCR, normalized to β-actin, and expressed as 2^−ΔΔCt^ relative to uninfected controls. Statistical significance was determined using two-way ANOVA followed by Tukey’s multiple comparisons. * *p* < 0.05 vs. control; # *p* < 0.05 vs. polybacterial infection. Data are presented as mean ± SEM of three independent biological replicates and two technical replicates. Abbreviations: Inf = infection; SEP = L-Sepiapterin; CDDO-Me = bardoxolone methyl; PI3K = phosphoinositide 3-kinase; AKT = protein kinase B; GLUT1 = glucose transporter 1; IRS = insulin receptor substrate; PEA-15 = phosphoprotein enriched in astrocytes-15.

**Figure 7 cells-14-01777-f007:**
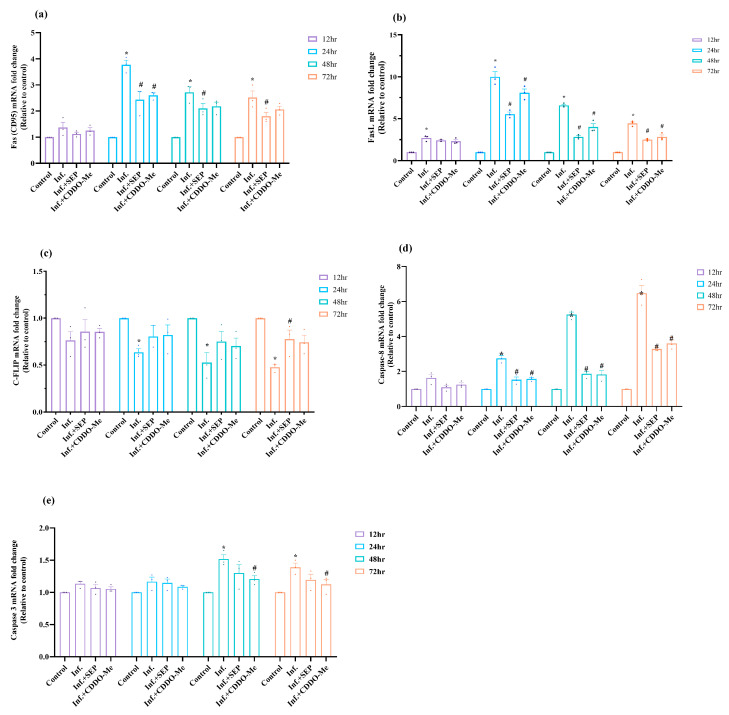
Polybacterial infection induces endothelial apoptosis via the Fas–caspase pathway, attenuated by L-Sepiapterin and CDDO-Me. HUVECs were infected and treated as above. (**a**,**b**) FAS and FASL mRNA expression. (**c**) mRNA expression of c-FLIP. (**d**,**e**) Caspase-8 and caspase-3 mRNA expression, respectively. mRNA levels were quantified by qRT-PCR, normalized to β-actin, and expressed as 2^−ΔΔCt^ relative to uninfected controls. Statistical significance was determined using two-way ANOVA followed by Tukey’s multiple comparisons. * *p* < 0.05 vs. control; # *p* < 0.05 vs. polybacterial infection. Data are presented as mean ± SEM of three independent biological replicates and two technical replicates. Abbreviations: Inf = infection; SEP = L-Sepiapterin; CDDO-Me = bardoxolone methyl; FAS = Fas receptor; FASL = Fas ligand; c-FLIP = cellular FLICE-like inhibitory protein.

**Figure 8 cells-14-01777-f008:**
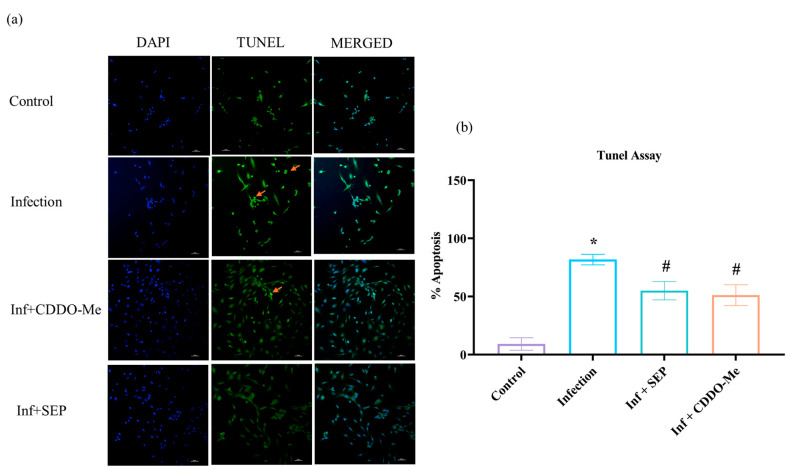
TUNEL and DAPI staining for detection of apoptotic cells. HUVECs were infected and treated as above (Section 2.9). (**a**) Representative images with 20× magnification showing dual staining revealed apoptotic nuclei characterized by DNA fragmentation (TUNEL-positive = bright green indicated in orange arrow) and nuclear morphology (DAPI, blue). (**b**) Bar graph shows the mean ± SEM of apoptotic nuclei quantified from six randomly selected fields per sample across three independent experiments (*n* = 3) and expressed as the percentage of total cells counted. Statistical significance was determined using two-way ANOVA followed by Tukey’s multiple comparisons. * *p* < 0.05 vs. control; # *p* < 0.05 vs. polybacterial infection. Scale bar 50 µm. Keys: Inf., infection with polybacteria; SEP, L-Sepiapterin; CDDO-Me, bardoxolone methyl.

**Figure 9 cells-14-01777-f009:**
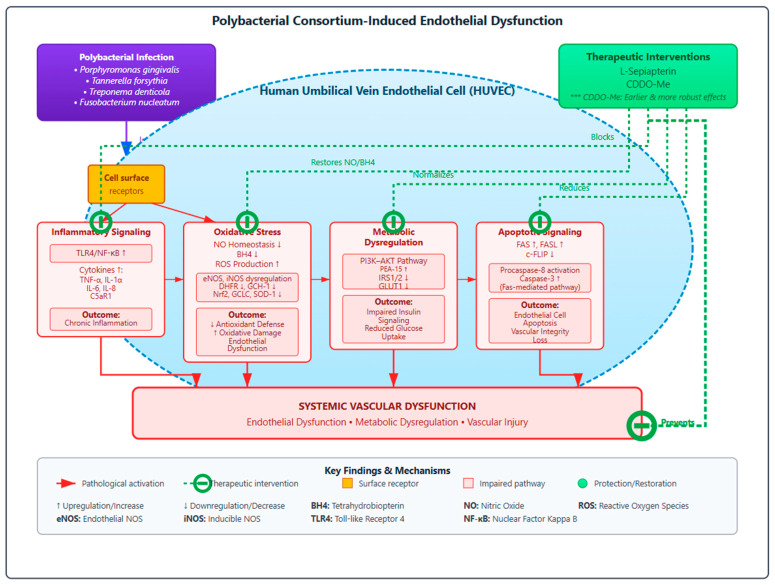
Schematic representation of periodontal polybacterial infection-induced endothelial dysfunction and its protection by L-Sepiapterin and CDDO-Me. Polybacterial infection disrupts endothelial homeostasis through interconnected mechanisms involving oxidative stress, inflammation, metabolic dysregulation, and apoptosis. Therapeutic modulation with L-Sepiapterin and CDDO-Me restores redox balance and suppresses inflammatory signaling, thereby preserving endothelial integrity. Key: *** CDDO-Me effectively suppressed inflammation and reduced apoptosis, collectively leading to an early and robust reversal of endothelial injury as compared to L-Sepiapterin. The schematic illustration was created using Figma (www.figma.com).

**Table 1 cells-14-01777-t001:** Primers used for quantitative real-time PCR.

Biomarkers	Forward Sequence (5′-3′)	Reverse Sequence (5′-3′)
eNOS	GAACCTGTGTGACCCTCACC	TGGCTAGCTGGTAACTGTGC
iNOS	GCCACAGAAGAGCCTGAGAG	GATCTCTGTGGGCGTGTGAT
DHFR	TCGACCATTGAACTGCATCGTCGCC	GGAATGGAGAACCAGGTTTTCCTACC
GCH-1	GAGCATCACCTTGTTCCATTTG	GCCAAGTTTACTGAGACCAAGGA
NRF2	CCCAATTCAGCCAGCCCAGC	AACGGGAATGTCTGCGCCAA
GCLC	ATGTGGACACCCGATGCAGTATT	TGTCTTGCTTGTAGTCAGGATGGTTT
SOD-1	CGGTTGAGATAGACAGG	TTAAGTGGTCTTGCACTCG
TLR4	ATATTGACAGGAAACCCCATCCA	AGAGAGATTGAGTAGGGGCATTT
NF-κB (p50)	GCAGCACTACTTCTTGACCACC	TCTGCTCCTGAGCATTGACGTC
TNF-a	GAGGCCAAGCCCTGGTATG	CGGGCCGATTGATCTCAGC
IL 1*α*	CCAGGCGTAGGTCTGGAGTCTCACTTGTCT	TGTTGCGGCAGGAAGGCTTAGGTATTATTC
IL-6	TCAATGAGGAGACTTGCCTG	GATGAGTTGTCATGTCCTGC
IL-8	ACTCCAAACCTTTCCACCCC	TTCTCAGCCCTCTTCAAAAACTTC
C5aR1	GCCCAGGAGACCAGAACAT	TATCCACAGGGGTGTTGAGG
PI3K	TGTGGAGCTCGCTAAAGTCA	CACTCCTGCCCTAAATGGGA
AKT	CTTTCGGCAAGGTGATCCTG	GTACTTCAGGGCTGTGAGGA
IRS-1	CGGAGAGCGATGGCTTCTC	GTTTGTGCATGCTCTTGGGTTT
IRS-2	ACAAGCCGCTGATTAATGAGGC	TGACTCGGCGTTACGCAGGCAC
GLUT-1	TTGCAGGCTTCTCCAACTGGAC	CAGAACCAGGAGCACAGTGAAG
PEA15	CCAGCGAAAAGAGTGAGGAGATC	TGTGCTCAATGTAGGAGAGGTTG
CASPASE-8	AGAAGAGGGTCATCCTGGGAGA	TCAGGACTTCCTTCAAGGCTGC
CASPASE-3	CATACTCCACAGCACCTGGTT	TCAAGCTTGTCGGCATACTGT
c-FLIP	GAC CCT TGT GCT TCC CTA	GTT AAT CAC ATG GAA CAA TTT CC
FAS (CD95)	TCC TCC AGG TGA AAG GAA AGC TAG G	AGA TTG TGT GAT GAA GGA CAT GGC
FASL	GCAGCCCTTCAATTACCCAT	CAGAGGTTGGACAGGGAAGAA
β-ACTIN	CGCGAGAAGATGACCCAGAT	AGCACAGCCTGGATAGCAAC

## Data Availability

The raw data supporting the conclusions of this article will be made available by the authors on request.

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
