# Peer review of "Role of Nitric Oxide and Nrf2 to Counteract Vascular Endothelial Dysfunction Induced by Periodontal Pathogens Using HUVECs"

_cells, 2025, doi:10.3390/cells14221777_

Round 1

Reviewer 1 Report

Comments and Suggestions for Authors

The authors of this study state in the INTRODUCTION that periodontal disease leads to inflammation, ROS, endothelial dysfunction, etc. They then go to show in an in vitro model that these are the mechanisms associating periodontal disease with endothelial dysfunction and presumably with clinical CVD. So, the authors are demonstrating what is already known, no? The numerous references in the bibliography suggest that quite a lot of knowledge regarding periodontal disease is known.  Please answer this query. The authors should state what is not known re periodontal disease and endothelial function and what knowledge gap their study is filling.

Technically the study is well done and well-written 

Author Response

Response to Reviewer 1 Comments

  1. Summary

Thank you very much for taking the time to review this manuscript. Please find detailed responses below and the corresponding revisions/corrections highlighted/in track changes in the re-submitted files.

  1. Point-by-point response to Comments and Suggestions for Authors

Comment 1: The authors of this study state in the INTRODUCTION that periodontal disease leads to inflammation, ROS, endothelial dysfunction, etc. They then go to show in an in vitro model that these are the mechanisms associating periodontal disease with endothelial dysfunction and presumably with clinical CVD. So, the authors are demonstrating what is already known, no? The numerous references in the bibliography suggest that quite a lot of knowledge regarding periodontal disease is known.  Please answer this query. The authors should state what is not known re periodontal disease and endothelial function and what knowledge gap their study is filling.

Technically the study is well done and well-written 

Response 1: Thank you for your constructive comments. We agreed with the comments and revised paragraph Page 3, line 102-125 of the revised manuscript.

The paragraph changed as “Although, the association between PD and endothelial dysfunction is well documented [12], the specific molecular mechanisms underlying polybacterial infection induced endothelial cells dysfunction remain underexplored. Our previous work showed that polybacterial infection markedly impairs NO synthesis and alters oxidative and inflammatory biomarkers. The findings highlighted the heightened vascular risk posed by polybacterial periodontal infection compared to single-species exposure [10]. However, the potential of redox-regulating agents such as L-sepiapterin and CDDO-Me to restore endothelial homeostasis under these conditions has not been studied. Therefore, this study addresses a critical knowledge gap by elucidating how these agents modulate the eNOS/BH4/NO axis, Nrf2–ARE pathway, and PI3K–AKT–IRS–GLUT signaling to counteract infection-induced oxidative stress, inflammation, and impaired metabolic signaling in endothelial cells. We hypothesize that polymicrobial infection disrupts endothelial function through reduced NO bioavailability, oxidative stress, inflammation, and impaired metabolic signaling, whereas treatment with either agent would restore endothelial integrity by re-establishing eNOS/BH4/NO coupling, activating Nrf2-mediated antioxidant defenses, and normalizing PI3K–AKT–IRS–GLUT signaling. In this study, we investigated the effects of polybacterial periodontal infection and evaluated the mechanistic potential of either L-sepiapterin or CDDO-Me as targeted therapeutic interventions using Human umbilical vein endothelial cells (HUVECs). The HUVECs were chosen as a model because they are a well-established and physiologically relevant system for studying vascular endothelial function. HUVECs exhibit key characteristics of systemic endothelial cells, including NO production, inflammatory cytokine responses, and regulation. They are widely used to investigate mechanisms of endothelial dysfunction, oxidative stress, and inflammation associated with cardiovascular and metabolic diseases [21, 22]”.

Reviewer 2 Report

Comments and Suggestions for Authors

This is a well-written, technically detailed manuscript exploring the mechanisms by which periodontal polybacterial infections disrupt endothelial function and how L-sepiapterin and bardoxolone methyl (CDDO-Me) may offer protection. The study is methodologically rigorous and addresses an important and understudied intersection between oral and systemic vascular health. However, some  refinements are required before publication.

1) While the mechanistic integration is strong, some of the findings largely confirm previously reported concepts (e.g., Nrf2 activation and eNOS uncoupling). Novelty lies mainly in using both L-sepiapterin and CDDO-Me together in the HoxBan co-culture system—this could be emphasized more explicitly.

2) Strengthen the “gap statement” in the introduction to highlight exactly what is new compared to prior endothelial infection studies.

3) Some methodological details need clarification:

- How were bacterial loads controlled or monitored during co-culture? Was bacterial viability  checked after 72 h?

- Were any bacterial viability controls used to ensure consistent MOI?

- Cytotoxicity assay duration and normalization method could be better described (e.g., baseline correction).

- Most data are at mRNA level—protein validation (Western blot, ELISA for cytokines or caspases) would greatly strengthen conclusions.

- Statistical treatment: n = 3 biological replicates may be insufficient for high variability in cell-based systems. Consider stating exact biological replicate numbers and confirm reproducibility.

Author Response

Response to Reviewer 2 Comments

  1. Summary

This is a well-written, technically detailed manuscript exploring the mechanisms by which periodontal polybacterial infections disrupt endothelial function and how L-sepiapterin and bardoxolone methyl (CDDO-Me) may offer protection. The study is methodologically rigorous and addresses an important and understudied intersection between oral and systemic vascular health. However, some refinements are required before publication.

Thank you very much for taking the time to review this manuscript. Please find detailed responses below and the corresponding revisions/corrections highlighted/in track changes in the re-submitted files.

  1. Point-by-point response to Comments and Suggestions for Authors

Comment 1) While the mechanistic integration is strong, some of the findings largely confirm previously reported concepts (e.g., Nrf2 activation and eNOS uncoupling). Novelty lies mainly in using both L-sepiapterin and CDDO-Me together in the HoxBan co-culture system—this could be emphasized more explicitly.

Response 1: Thank you for your comments. We agreed with the comments and revised accordingly. Page 3-4, line 102-125 of the revised manuscript.

Comment 2) Strengthen the “gap statement” in the introduction to highlight exactly what is new compared to prior endothelial infection studies.

Response 2: Thank you for your constructive comments. We have now revised paragraph Page 3-4, line 102-125 of the revised manuscript.

Comment 3) Some methodological details need clarification: - How were bacterial loads controlled or monitored during co-culture? Was bacterial viability checked after 72 h?

Response 3: Thank you for your comments. Now we have included the MOI used for the HoxBan coculture. Since in Hoxban coculture method the bacteria could interact with the cells at the interface of the agar surface and the coverslip. We cannot assure the exact MOI at this interface. Bacteria in the dipper of the media may not interact with the cells. Here, the MOI is an estimate number. The bacteria grow into a biofilm, and the HUVECs are placed on top. The actual number of bacteria the endothelial cells contact may not be uniform. This is the limitation of this technique, and we have now mentioned the limitation of the technique in the manuscript Page 23 line 748-757 in the revised manuscript. We also conducted experiments to ensure bacterial viability through the HoxBan experiment. Now we have described this in the methods section in the revised manuscript. Page 23 line 157-193.

Comment 4) - Were any bacterial viability controls used to ensure consistent MOI?

Response 4: Response 4: Thank you for your comments. We have included the MOI information and additional details of the HoxBan experiment in the Methods section, as also noted in Response 3.

Comment 5)- Cytotoxicity assay duration and normalization method could be better described (e.g., baseline correction).

Response 5: We thank the reviewer for this comment. The MTT assay was performed at multiple time points (6–72 h) to assess the cytotoxicity of L-sepiapterin and CDDO-Me in HUVECs. Cell viability was calculated by normalizing the absorbance of treated wells to that of the corresponding control wells incubated with complete vascular basal media, with baseline correction. The description of method has been added in the revised methods section. Page 6, line 203-206 of the revised manuscript.

Comment 6) - Most data are at mRNA level—protein validation (Western blot, ELISA for cytokines or caspases) would greatly strengthen conclusions.

Response 6: Thank you for your comments and suggestions. We measured gene expression levels along with BH4 (ELISA), NO (Griess reaction), and ROS (luminescent assay) in culture supernatants. These results aligned with gene expression patterns of NO-related (eNOS, iNOS) and oxidative stress markers (Nrf2, GCLC, SOD1), indirectly supporting our findings. However, protein levels for all biomarkers were not measured, which is a limitation of this study. We plan to include these assays in future work. This limitation has been added to the revised manuscript (page 24, line 751-759).

Comment 7)- Statistical treatment: n = 3 biological replicates may be insufficient for high variability in cell-based systems. Consider stating exact biological replicate numbers and confirm reproducibility.

Response 7: Thank you for your constructive comments. We performed three biological replicates and two technical replicates. We have stated in the methods section as well as figure legends in the revised manuscript.

Reviewer 3 Report

Comments and Suggestions for Authors

This is a scientifically sound study that investigates an important and emerging topic: the mechanistic interplay between periodontal polybacterial infection and vascular endothelial dysfunction. Using HUVECs, the authors provide convincing experimental evidence that red-complex and orange-complex bacteria disrupt nitric oxide signalling, inducing oxidative stress, inflammation and apoptosis, as well as impairing metabolic pathways.

I have made some suggestions on how you could improve your work. This doesn't mean that you have to agree with them or rewrite your work in the same way. They are just suggestions to help you see things from a different perspective.

1) Although the "Introduction" section is strong, please briefly explain why HUVECs were chosen as a model (e.g. their relevance to systemic endothelial physiology and vascular inflammation).

2) Also include, please, a brief note on the clinical significance or perspective. For example, you could link PD-related vascular dysfunction to specific outcomes such as atherosclerosis, hypertension or insulin resistance.

3) The authors reference earlier work in pHAECs (lines 86-88). The rationale would be strengthened by specifying what new mechanisms or signalling interactions are explored here (e.g. Nrf2-PI3K-AKT crosstalk, dual pharmacological intervention).

4) Please, streamline redundant phrases. Example: The phrase "periodontal disease (PD)" is defined twice (lines 38 and 44). The second instance could be shortened for conciseness.

5) Consider mentioning the potential synergistic or comparative effects of L-sepiapterin and CDDO-Me in the hypothesis to strengthen the rationale for dual testing.

6) Please, add a graphical abstract or pathway schematic illustrating the hypothesised mechanism (NO-BH4-eNOS and Nrf2-NF-κB-PI3K-AKT axes). This would provide a visual summary of the complex molecular interactions.

7) Although the individual assays are well described, the overall experimental workflow and group allocation (controls, infected, treated and infection + treatment) could be summarised more explicitly. Please, add a short paragraph or schematic overview at the start of Section 2 summarising the experimental groups and time points. This will help readers to grasp the experimental design before they dive into the individual methods.

8) While the HoxBan method is described, the exact multiplicity of infection used in the co-culture system is inconsistent across the subsections (e.g. the MOI is only mentioned in the TUNEL assay). Please, clarify the calculated MOI in subsection 2.3 (HoxBan co-culture) and confirm bacterial viability or anaerobic maintenance throughout co-culture.

9) Although the text mentions duplicates or triplicates, it is unclear whether the data were averaged per biological replicate or pooled from technical replicates. Please, clarify whether biological or technical replicates were used.

10) RNA quality assessment is limited to NanoDrop absorbance ratios. Please, indicate whether RNA integrity (RIN) was verified, especially for the reliability of quantitative PCR. Software versions for imaging and qPCR analysis should be added.

11) The primers are listed, but there is no mention of validation. Please, add a sentence noting the verification of specificity.

12) Although β-actin is used as a reference gene, its stability under infection and inflammatory conditions should be justified.

13) The microscopy and quantification methods are sound, but the image analysis parameters (e.g. thresholding or software criteria) are not described.

14) The description lacks details on post hoc tests and normality checks.

15) In the "Results" section, please, add a brief introductory paragraph. Currently, the section launches directly into data without orienting the reader. Add a short overview (3-4 lines) at the beginning of Section 3, summarising the content of the "Results" section.

16) Each subsection in the "Results" section is descriptive, but lacks integrative sentences summarising the magnitude or hierarchy of the effects. Please add one or two sentences that summarise the main pattern (e.g. "CDDO-Me exhibited an earlier but transient effect, whereas L-Sepiapterin acted more gradually yet persistently"). This improves readability and facilitates comparison between treatments.

17) Some parts mix fold-change descriptions with p-values, but do not always specify the reference group or the type of variance. Please, ensure that each comparison specifies the reference group (control or infection) and confirm consistent reporting of mean ± SEM.

18) The "Results" section contains six figures, each focusing on a single signalling cluster. While this approach is scientifically justified, readers may find it fragmented. Please, add Figure 8, which should be a schematic summary or table integrating all observed changes (increase or decrease) in each gene and protein across conditions and time points. This would help synthesise the large dataset visually.

19) All results are based on mRNA data and biochemical assays (NO, BH4 and ROS). There is no mention of Western blot or ELISA validation of protein expression or phosphorylation. If these data exist, include them briefly; if not, note this as a limitation in the "Discussion" section.

20) Some paragraphs contain repetitive descriptions ("infection significantly increased..", "both treatments reduced.."), which are repeated in every subsection. These can be shortened to make the text easier to read.

21) To strengthen the mechanistic argument, the authors could correlate: NO vs. ROS or Nrf2 vs. caspase-3 expression. This would quantify the relationships between oxidative stress, inflammation and apoptosis.

22) Please, add short linking sentences between the subsections in the 'Results' section.

23) The "Discussion" section jumps straight into mechanistic detail. Readers would benefit from an initial overview summarising the study's main findings before the interpretation begins. Please, add a short introductory paragraph (three to four sentences) summarising the major outcomes and their novelty.

24) Several subsections (particularly lines 505-520, 549-576 and 597-610) contain explanations that are similar to those in the "Results" section. Redundant sentences should be condensed (e.g. repeated mentions of "disruption of NO metabolism" or '"activation of NF-κB"), and the focus should be shifted towards interpretation, implications, and integration across pathways. Rather than restating that infection "induces oxidative stress, inflammation, and apoptosis' in each paragraph, provide a summary at the end: 'These cumulative effects underscore that endothelial injury results from an interconnected network of oxidative, inflammatory, and metabolic disturbances rather than isolated mechanisms."

25) The discussion remains primarily at the cellular level. It would benefit from linking endothelial dysfunction to clinical outcomes such as atherosclerosis, insulin resistance and cardiovascular complications in patients with periodontitis. Please, add one paragraph discussing practical implications.

26) The discussion lacks a clear limitations paragraph. Please, include a concise paragraph acknowledging methodological constraints.

27) The text concludes with a description of Figure 8 without providing a clear forward-looking statement. Please, conclude the discussion by linking the basic findings to translational research.

28) Line 497 claims that L-sepiapterin and CDDO-Me "synergistically restore endothelial homeostasis", but the experimental design tested them individually, not in combination. Replace "synergistically" with "complementarily" or "via distinct yet convergent mechanisms".

29) The discussion is long and lacks internal structure. Please, introduce logical sub-sections (or at least transition phrases): Nitric oxide and BH4 metabolism, Nrf2-mediated redox regulation, Inflammatory and complement pathways, Metabolic and apoptotic dysregulation, Integrated pathogenic model and therapeutic implications. This improves readability and mirrors the results structure.

30) In the "Conclusions" section, mentioning the extent or magnitude of improvement (e.g. "significantly restored NO levels" or "markedly reduced ROS production") would provide stronger closure and emphasise the robustness of the results. Stating explicitly how this study advances current understanding would also increase the scientific significance.

31) While the need for animal and clinical validation is acknowledged, including a sentence that links these findings to potential periodontal–cardiovascular crosstalk mechanisms or adjunctive therapeutic strategies would enhance the study's relevance.

While this manuscript presents valuable findings, improving the clarity and depth of the discussion, as well as the level of methodological detail, would enhance its impact even further. The findings are well supported by the experimental approaches employed. I recommend accepting the manuscript with some revisions.

Author Response

Response to Reviewer 3 Comments

  1. Summary

This is a scientifically sound study that investigates an important and emerging topic: the mechanistic interplay between periodontal polybacterial infection and vascular endothelial dysfunction. Using HUVECs, the authors provide convincing experimental evidence that red-complex and orange-complex bacteria disrupt nitric oxide signalling, inducing oxidative stress, inflammation and apoptosis, as well as impairing metabolic pathways.

I have made some suggestions on how you could improve your work. This doesn't mean that you have to agree with them or rewrite your work in the same way. They are just suggestions to help you see things from a different perspective.

Response: Thank you very much for taking the time to review this manuscript. Please, find detailed responses below and the corresponding revisions/corrections highlighted/in track changes in the re-submitted manuscript files.

  1. Point-by-point response to Comments and Suggestions for Authors

Comment 1) Although the "Introduction" section is strong, please briefly explain why HUVECs were chosen as a model (e.g. their relevance to systemic endothelial physiology and vascular inflammation).

Response 1: Thank you for your valuable comment. We agree with the suggestion and have revised the introduction accordingly. The changes are revealed on page 3, line 120-123 of the revised manuscript.

Comment 2) Also include, please, a brief note on the clinical significance or perspective. For example, you could link PD-related vascular dysfunction to specific outcomes such as atherosclerosis, hypertension or insulin resistance.

Response 2: Thank you for your comments. We agreed with the comments and revised the introduction section accordingly. Page 3, line 82-85 of the revised manuscript.

Comment 3) The authors reference earlier work in pHAECs (lines 86-88). The rationale would be strengthened by specifying what new mechanisms or signalling interactions are explored here (e.g. Nrf2-PI3K-AKT crosstalk, dual pharmacological intervention).

Response 3: Thank you for your valuable comments. We have revised the paragraph accordingly and incorporated the pharmacological intervention details on page 3, line 102-125 of the revised manuscript.

Comment 4) Please, streamline redundant phrases. Example: The phrase "periodontal disease (PD)" is defined twice (lines 38 and 44). The second instance could be shortened for conciseness.

Response 4: Thank you for your comments. We apologize for the oversight. The sentence has now been corrected on pages 3, line 59 of the revised manuscript.

Comment 5) Consider mentioning the potential synergistic or comparative effects of L-sepiapterin and CDDO-Me in the hypothesis to strengthen the rationale for dual testing.

Response 5: Thank you for your valuable comment. In this study, we employed L-sepiapterin as a nitric oxide (NO) precursor and CDDO-Me as an Nrf2 pathway activator. These pharmacological agents were evaluated independently to delineate their specific mechanistic effects. The synergistic effects of both interventions were not assessed in this study. The manuscript has been revised accordingly (Page 3, lines 82-125 of the revised manuscript).

Comment 6) Please, add a graphical abstract or pathway schematic illustrating the hypothesised mechanism (NO-BH4-eNOS and Nrf2-NF-κB-PI3K-AKT axes). This would provide a visual summary of the complex molecular interactions.

Response 6: Thank you for your insightful comment. We have now included a graphical abstract illustrating the hypothesized mechanisms. The graphical abstract has been added to the revised manuscript (Page 2).

Comment 7) Although the individual assays are well described, the overall experimental workflow and group allocation (controls, infected, treated and infection + treatment) could be summarised more explicitly. Please, add a short paragraph or schematic overview at the start of Section 2 summarising the experimental groups and time points. This will help readers to grasp the experimental design before they dive into the individual methods.

Response 7: Thank you for your comment. As per your suggestion We have now included a figure for Schematic overview of the experimental design has been added to the revised manuscript (Page 4, Figure 01).

Comment 8) While the HoxBan method is described, the exact multiplicity of infection used in the co-culture system is inconsistent across the subsections (e.g. the MOI is only mentioned in the TUNEL assay). Please, clarify the calculated MOI in subsection 2.3 (HoxBan co-culture) and confirm bacterial viability or anaerobic maintenance throughout co-culture.

Response 8: Thank you for your comments. Now we have included the MOI used for the HoxBan coculture. Since in Hoxban coculture method the bacteria could interact with the cells at the interface of the agar surface and the coverslip. We cannot assure the exact MOI at this interface. Bacteria in the dipper of the media may not interact with the cells. Here, the MOI is an estimate number. The bacteria grow into a biofilm, and the HUVECs are placed on top. The actual number of bacteria the endothelial cells contact may not be uniform. This is the limitation of this technique, and we have now mentioned the limitation of the technique in the manuscript Page 23 line 748-757 in the revised manuscript. We also conducted experiments to ensure bacterial viability through the HoxBan experiment. Now we have described this in the methods section in the revised manuscript. Page 23 line 157-193.

Comment 9) Although the text mentions duplicates or triplicates, it is unclear whether the data were averaged per biological replicate or pooled from technical replicates. Please, clarify whether biological or technical replicates were used.

Response 9: Thank you for your comments. We performed three biological replicates and two technical replicates. We have stated the biological replicates in the methods section as well as figure legends in the revised manuscript.

Comment 10) RNA quality assessment is limited to NanoDrop absorbance ratios. Please, indicate whether RNA integrity (RIN) was verified, especially for the reliability of quantitative PCR. Software versions for imaging and qPCR analysis should be added.

Response 10: We thank the reviewer for this valuable comment. RNA purity was assessed using NanoDrop spectrophotometry by evaluating A260/A280 and A260/A230 ratios, and only samples with acceptable purity values were used for cDNA synthesis. Although RNA integrity number (RIN) values were not determined, the specificity and reliability of qPCR reactions were verified through melt curve analysis, which confirmed the amplification of single, specific products without primer-dimer formation. Additionally, the software details have been added to the revised manuscript: qPCR data were analyzed using Bio-Rad CFX Maestro Version 2. We have revised the methods section and acknowledged the limitation of revised manuscript.

Comment 11) The primers are listed, but there is no mention of validation. Please, add a sentence noting the verification of specificity.

Response 11: Thank you for your comment. We used previously validated primers and confirmed their specificity through melting curve analysis. We added the information in the Page 7 Line 260-261.

Comment 12) Although β-actin is used as a reference gene, its stability under infection and inflammatory conditions should be justified.

Response 12: We thank the reviewer for this insightful comment. β-actin gene was selected as the reference for the present study. Our preliminary analysis showed minimal variation in β-actin Ct values, consistent with previous studies validating its use as a reference gene in endothelial cells under polybacterial infection and oxidative stress condition. We have included the references with this response [1, 2].

Comment 13) The microscopy and quantification methods are sound, but the image analysis parameters (e.g. thresholding or software criteria) are not described.

Response 13: Thank you for your comment. As per your suggestion We have now included the image analysis parameters (e.g. thresholding or software criteria) in the methods section. (Page 8, Line 285-294).

Comment 14) The description lacks details on post hoc tests and normality checks.

Response 14: We appreciate the reviewer’s comment. Data normality was assessed using the Shapiro–Wilk test, and Q–Q plots were used for visual inspection of distribution. Depending on the comparison, either Student’s t-test or two-way ANOVA was applied, followed by Tukey’s multiple comparisons post hoc test to determine group-wise significance. We have revised the manuscript accordingly. Page 8, Line 296-302.

Comment 15) In the "Results" section, please, add a brief introductory paragraph. Currently, the section launches directly into data without orienting the reader. Add a short overview (3-4 lines) at the beginning of Section 3, summarising the content of the "Results" section.

Response 15: We thank the reviewer for this suggestion. A brief introductory paragraph has been added at the beginning of the results section of the revised manuscript.

Comment 16) Each subsection in the "Results" section is descriptive, but lacks integrative sentences summarising the magnitude or hierarchy of the effects. Please add one or two sentences that summarise the main pattern (e.g. "CDDO-Me exhibited an earlier but transient effect, whereas L-Sepiapterin acted more gradually yet persistently"). This improves readability and facilitates comparison between treatments.

Response 16: We thank the reviewer for this suggestion. We have added integrative summary sentences at the end of each subsection in the results in the revised manuscript.

Comment 17) Some parts mix fold-change descriptions with p-values, but do not always specify the reference group or the type of variance. Please, ensure that each comparison specifies the reference group (control or infection) and confirm consistent reporting of mean ± SEM.

Response 17: We appreciate the reviewer’s observation. We have revised the results section to clearly specify the reference group (control or infection) and statistical comparisons in the revised manuscript.

Comment 18) The "Results" section contains six figures, each focusing on a single signalling cluster. While this approach is scientifically justified, readers may find it fragmented. Please, add Figure 8, which should be a schematic summary or table integrating all observed changes (increase or decrease) in each gene and protein across conditions and time points. This would help synthesise the large dataset visually.

Response 18: Thank you for the suggestion. A schematic summary has already been included as Figure 9 at the end of the discussion section, summarizing the observed changes in key genes analyzed. The figure illustrates the polybacterial infection disrupts endothelial homeostasis through oxidative stress, inflammation, metabolic imbalance, and apoptosis, and L-sepiapterin and CDDO-Me restore endothelial integrity. We would appreciate your advice on whether this figure should remain in the discussion section or be moved to the results section.

Comment 19) All results are based on mRNA data and biochemical assays (NO, BH4 and ROS). There is no mention of Western blot or ELISA validation of protein expression or phosphorylation. If these data exist, include them briefly; if not, note this as a limitation in the "Discussion" section.

Response 19: Thank you for your comments and suggestions. In this study, we focused on gene expression and biochemical assays to assess endothelial responses. Protein-level validation was not performed, which is one of the limitations of present study. We have acknowledged this limitation in the revised manuscript (page no.23 Line 751-752) and plan to include protein expression and phosphorylation analyses in future studies.

Comment 20) Some paragraphs contain repetitive descriptions ("infection significantly increased..", "both treatments reduced.."), which are repeated in every subsection. These can be shortened to make the text easier to read.

Response 20: Thank you for the valuable feedback. We have carefully revised the “Results” section to minimize repetitive phrasing in the revised manuscript.

Comment 21) To strengthen the mechanistic argument, the authors could correlate: NO vs. ROS or Nrf2 vs. caspase-3 expression. This would quantify the relationships between oxidative stress, inflammation and apoptosis.

Response 21: We thank the reviewer for this suggestion. In the revised manuscript, we have addressed this point in the discussion section by correlating NO versus ROS levels and Nrf2 versus caspase-3 expression, highlighting the mechanistic links between oxidative stress, inflammation, and apoptosis. Page 20-23.

Comment 22) Please, add short linking sentences between the subsections in the 'Results' section.

Response 22: We thank the reviewer for this helpful suggestion. Short linking sentences have been added at the beginning of each results subsection.

Comment 23) The "Discussion" section jumps straight into mechanistic detail. Readers would benefit from an initial overview summarising the study's main findings before the interpretation begins. Please, add a short introductory paragraph (three to four sentences) summarising the major outcomes and their novelty.

Response 23: We thank the reviewer for this valuable suggestion. In response, we have added a brief introductory paragraph at the beginning of the discussion section summarizing the major outcomes and novelty of the study. Page 20, Line 580-595

Comment 24) Several subsections (particularly lines 505-520, 549-576 and 597-610) contain explanations that are similar to those in the "Results" section. Redundant sentences should be condensed (e.g. repeated mentions of "disruption of NO metabolism" or '"activation of NF-κB"), and the focus should be shifted towards interpretation, implications, and integration across pathways. Rather than restating that infection "induces oxidative stress, inflammation, and apoptosis' in each paragraph, provide a summary at the end: 'These cumulative effects underscore that endothelial injury results from an interconnected network of oxidative, inflammatory, and metabolic disturbances rather than isolated mechanisms."

Response 24: We appreciate the reviewer’s insightful comment. Redundant descriptions in the specified subsections have been condensed, and the discussion has been revised to emphasize interpretation and cross-pathway integration.

Comment 25) The discussion remains primarily at the cellular level. It would benefit from linking endothelial dysfunction to clinical outcomes such as atherosclerosis, insulin resistance and cardiovascular complications in patients with periodontitis. Please, add one paragraph discussing practical implications

Response 25: We thank the reviewer for this valuable suggestion. A new paragraph has been added to the discussion linking endothelial dysfunction observed in our study to clinical outcomes. The revised section now discusses how chronic polybacterial infection–induced endothelial impairment may contribute to systemic conditions. Page 23 Line 724-749

Comment 26) The discussion lacks a clear limitations paragraph. Please, include a concise paragraph acknowledging methodological constraints.

Response 26: We appreciate the reviewer’s insightful comment. A concise paragraph outlining the study’s limitations has been added to discussion section 4.6. Page 23-24. Line 750-759

Comment 27) The text concludes with a description of Figure 8 without providing a clear forward-looking statement. Please, conclude the discussion by linking the basic findings to translational research

Response 27: Thank you for the helpful suggestion. A concluding paragraph has been added to the discussion to provide a forward-looking perspective. Now the figured number has changed to Figure 9.

Comment 28) Line 497 claims that L-sepiapterin and CDDO-Me "synergistically restore endothelial homeostasis", but the experimental design tested them individually, not in combination. Replace "synergistically" with "complementarily" or "via distinct yet convergent mechanisms".

Response 28: We appreciate the reviewer’s observation. As per reviewer’s suggestion and replaced the word synergistically with complementarily. Page 20, Line 593.

Comment 29) The discussion is long and lacks internal structure. Please, introduce logical sub-sections (or at least transition phrases): Nitric oxide and BH4 metabolism, Nrf2-mediated redox regulation, Inflammatory and complement pathways, Metabolic and apoptotic dysregulation, Integrated pathogenic model and therapeutic implications. This improves readability and mirrors the results structure.

Response 29: Thank you for the valuable recommendation. The discussion section has been reorganized and included subsections as per reviewer’s suggestions.

Comment 30) In the "Conclusions" section, mentioning the extent or magnitude of improvement (e.g. "significantly restored NO levels" or "markedly reduced ROS production") would provide stronger closure and emphasise the robustness of the results. Stating explicitly how this study advances current understanding would also increase the scientific significance.

Response 30: We thank the reviewer for the suggestion. The conclusions have been revised as per the reviewer’s suggestions. Page 24, Line 769-781

Comment 31) While the need for animal and clinical validation is acknowledged, including a sentence that links these findings to potential periodontal–cardiovascular crosstalk mechanisms or adjunctive therapeutic strategies would enhance the study's relevance.

Response 31: We thank the reviewer for this suggestion. A sentence has been added to the conclusions linking our findings to potential periodontal–cardiovascular crosstalk. Page 24, Line 769-781

While this manuscript presents valuable findings, improving the clarity and depth of the discussion, as well as the level of methodological detail, would enhance its impact even further. The findings are well supported by the experimental approaches employed. I recommend accepting the manuscript with some revisions.

Response: We thank the reviewer for the positive assessment and constructive feedback. We have addressed all suggested revisions, including enhancing the clarity and depth of the discussion, and interpretation of the results. We appreciate the reviewer’s recommendation and believe the revised manuscript is now strengthened in both scientific rigor and readability.

References:

  1. Sampath, C., et al., Polybacterial Periodontal Infection Alters oxidative and Inflammatory Biomarkers in Primary Human Aortic Endothelial Cell (pHAECs). J Pharm Pharmacol Res, 2025. 9(2): p. 45-53.
  2. Li, T., et al., Identification of suitable reference genes for real-time quantitative PCR analysis of hydrogen peroxide-treated human umbilical vein endothelial cells. BMC Mol Biol, 2017. 18(1): p. 10

Round 2

Reviewer 1 Report

Comments and Suggestions for Authors

the authors have answered my query satisfactorily